# cAMP−EPAC−PKCε−RIM1α signaling regulates presynaptic long-term potentiation and motor learning

Xin-Tai Wang[1,2†], Lin Zhou[1†], Bin-Bin Dong[1†], Fang-Xiao Xu[1], De-Juan Wang[1], En-Wei Shen[1], Xin-Yu Cai[1], Yin Wang[3], Na Wang[1], Sheng-Jian Ji[4], Wei Chen[1], Martijn Schonewille[5], J Julius Zhu[6*], Chris I De Zeeuw[5,7*], Ying Shen[1,8,9*]

[1]Department of Physiology and Department of Psychiatry, Sir Run Run Shaw Hospital, Zhejiang University School of Medicine, Hangzhou, China; [2]Institute of Life Sciences, College of Life and Environmental Sciences, Hangzhou Normal University, Hangzhou, China; [3]Key Laboratory of Cranial Cerebral Diseases, Department of Neurobiology of Basic Medical College, Ningxia Medical University, Yinchuan, China; [4]Department of Biology, Southern University of Science and Technology, Shenzhen, China; [5]Department of Neuroscience, Erasmus MC, Rotterdam, Netherlands; [6]Department of Pharmacology, University of Virginia, Charlottesville, United States; [7]Netherlands Institute for Neuroscience, Royal Academy of Sciences, Amsterdam, Netherlands; [8]International Institutes of Medicine, the Fourth Affiliated Hospital, Zhejiang University School of Medicine, Yiwu, China; [9]Key Laboratory of Medical Neurobiology of Zhejiang Province, Zhejiang University School of Medicine, Hangzhou, China

*For correspondence:
jjzhu@virginia.edu (JJuliusZ);
c.dezeeuw@erasmusmc.nl
(CIDZ);
yshen@zju.edu.cn (YS)

†These authors contributed
equally to this work

Competing interest: The authors
declare that no competing
interests exist.

Reviewing Editor: Jun Ding,
Stanford University, United
States

**Abstract** The cerebellum is involved in learning of fine motor skills, yet whether presynaptic plasticity contributes to such learning remains elusive. Here, we report that the EPAC-PKCε module has a critical role in a presynaptic form of long-term potentiation in the cerebellum and motor behavior in mice. Presynaptic cAMP−EPAC−PKCε signaling cascade induces a previously unidentified threonine phosphorylation of RIM1α, and thereby initiates the assembly of the Rab3A−RIM1α−Munc13-1 tripartite complex that facilitates docking and release of synaptic vesicles. Granule cell-specific blocking of EPAC−PKCε signaling abolishes presynaptic long-term potentiation at the parallel fiber to Purkinje cell synapses and impairs basic performance and learning of cerebellar motor behavior. These results unveil a functional relevance of presynaptic plasticity that is regulated through a novel signaling cascade, thereby enriching the spectrum of cerebellar learning mechanisms.

## Editor's evaluation

The cerebellum plays a critical role in motor learning, but exactly which forms of synaptic plasticity contribute to learning and the underlying molecular mechanisms remain poorly understood. In this study, Wang and colleagues show that presynaptic long-term potentiation at the parallel fiber to Purkinje cell synapse is required for one form of motor learning, and involves a previously-unknown signaling cascade, where EPAC activation leads to PKCε-dependent threonine phosphorylation of RIM1α. The evidence is compelling and convincing. This study provides fundamental and new insights into the underlying mechanisms and functional consequences of presynaptic LTP.

## Introduction

The cerebellum has historically been viewed as a motor coordination center (*Ito, 2005*). Recent evidence implicates that the cerebellum is also involved in a variety of learning-dependent high-level behaviors, including motor precision (*Wagner and Luo, 2020*; *De Zeeuw, 2021*) as well as cognitive and emotional functions (*Schmahmann et al., 2019*). The unique capability of the cerebellum to govern fine-tuned motor and cognitive skills at a high temporal resolution critically depends on delicate coordination of multiple forms of plasticity (*De Zeeuw, 2021*). Indeed, recent studies indicate that, in addition to the renowned postsynaptic long-term depression (LTD) (*Ito, 2005*) and long-term potentiation (LTP) (*Schonewille et al., 2010*), other forms of synaptic or non-synaptic plasticity may also contribute to cerebellar motor learning (*Raymond and Medina, 2018*; *De Zeeuw, 2021*). Relatively speaking, the molecular underpinnings of presynaptic plasticity in the cerebellar cortex are less understood (*Wang et al., 2021*), although early studies have shown that presynaptic Ca influx, Ca-sensitive adenylate cyclase, and cyclic adenosine monophosphate (cAMP) production are required for presynaptic LTP (*Byrne and Kandel, 1996*; *Salin et al., 1996*; *Storm et al., 1998*). Moreover, the function of presynaptic plasticity on cerebellar motor learning remains to be elucidated (*Le Guen and De Zeeuw, 2010*; *De Zeeuw, 2021*), although it was suggested that adenylyl cyclase-dependent LTP participates in rotarod learning (*Storm et al., 1998*).

In particular, the function of cAMP-dependent protein kinase A (PKA) on transmission release has been the subject of debate. *Lonart et al., 2003* found that RIM1α-Ser413 is phosphorylated by PKA, which is required for presynaptic LTP. However, the mice with dysfunctional RIM1α-Ser413 mutation exhibit normal presynaptic LTP in the cerebellum and the hippocampus (*Kaeser et al., 2008*; *Yang and Calakos, 2010*), questioning the role of RIM1α-Ser413 and PKA in presynaptic LTP. Thus, how RIM1α is activated during presynaptic plasticity needs to be revisited.

In this study, we identified a new presynaptic signaling module that comprises EPAC (exchange protein directly activated by cAMP) and PKCε (epsilon isozyme of protein kinase C). This signaling module controls threonine phosphorylation of RIM1α, initiates the assembly of a Rab3A-RIM1α-Munc13-1 tripartite complex, and thereby facilitates docking and release of synaptic vesicles at parallel fiber (PF) to Purkinje cell (PC) synapses, which is in line with previous work (*Martín et al., 2020*) showing β-adrenergic receptors/EPAC signaling modulates PF release using EPAC2 knockout mice. Importantly, presynaptic ablation of either EPAC or PKCε is sufficient to inhibit presynaptic LTP and impair motor performance and motor learning. These data unveil a new signaling cascade governing presynaptic LTP and demonstrate that presynaptic plasticity is essential to cerebellar motor learning.

## Results

### EPAC induces PKCε-dependent threonine phosphorylation of RIM1α

In order to study the function of EPACs at synapses, a series of centrifugations were employed to prepare cerebellar synaptosomes containing a number of synaptic proteins (*Figure 1A*). We found that most of EPAC1 and EPAC2 overlapped with vesicle glutamate transporter 1 (vGluT1) (*Figure 1B*), which is enriched at PF terminals (*Hioki et al., 2003*). Of the total synaptosomes, PF synapses (vGluT1$^+$EAAT4$^+$) and climbing fiber (CF) synapses (vGluT2$^+$EAAT4$^+$) constituted 88.8% and 7.5% of the total, respectively (*Figure 1—figure supplement 1*). Co-immunoprecipitation (co-IP) performed using synaptosomes showed that both EPAC1 and EPAC2 were precipitated by the RIM1 antibody (*Figure 1C*), indicating the ability of EPAC to interact with RIM1. To specify the action of EPAC, RIM1 was extracted from the synaptosomes by anti-RIM1 antibody-based co-IP (*Figure 1A*). Interestingly, we found that pan-phospho-threonine (p-Thr) antibodies detected only a weak signal in control synaptosomes, but a strong band in synaptosomes treated with 8-pCPT, a specific activator of EPAC (*Figure 1D*). In contrast, the level of pan-phospho-serine (p-Ser) remained unchanged after 8-pCPT treatment (*Figure 1D*). These results were confirmed by co-transfecting HA-RIM1α with Flag-EPAC1 or Flag-EPAC2 in HEK cells, where both types of EPAC as well as RIM1α were preferentially distributed along cell membrane (*Figure 1—figure supplement 2A*). Again, HA-RIM1α was precipitated with the HA antibody to characterize p-Ser and p-Thr of RIM1α. Consistent with in vivo assay, neither EPAC1 nor EPAC2 altered serine phosphorylation of RIM1α, but both increased phosphorylation of threonine (*Figure 1—figure supplement 2B*).

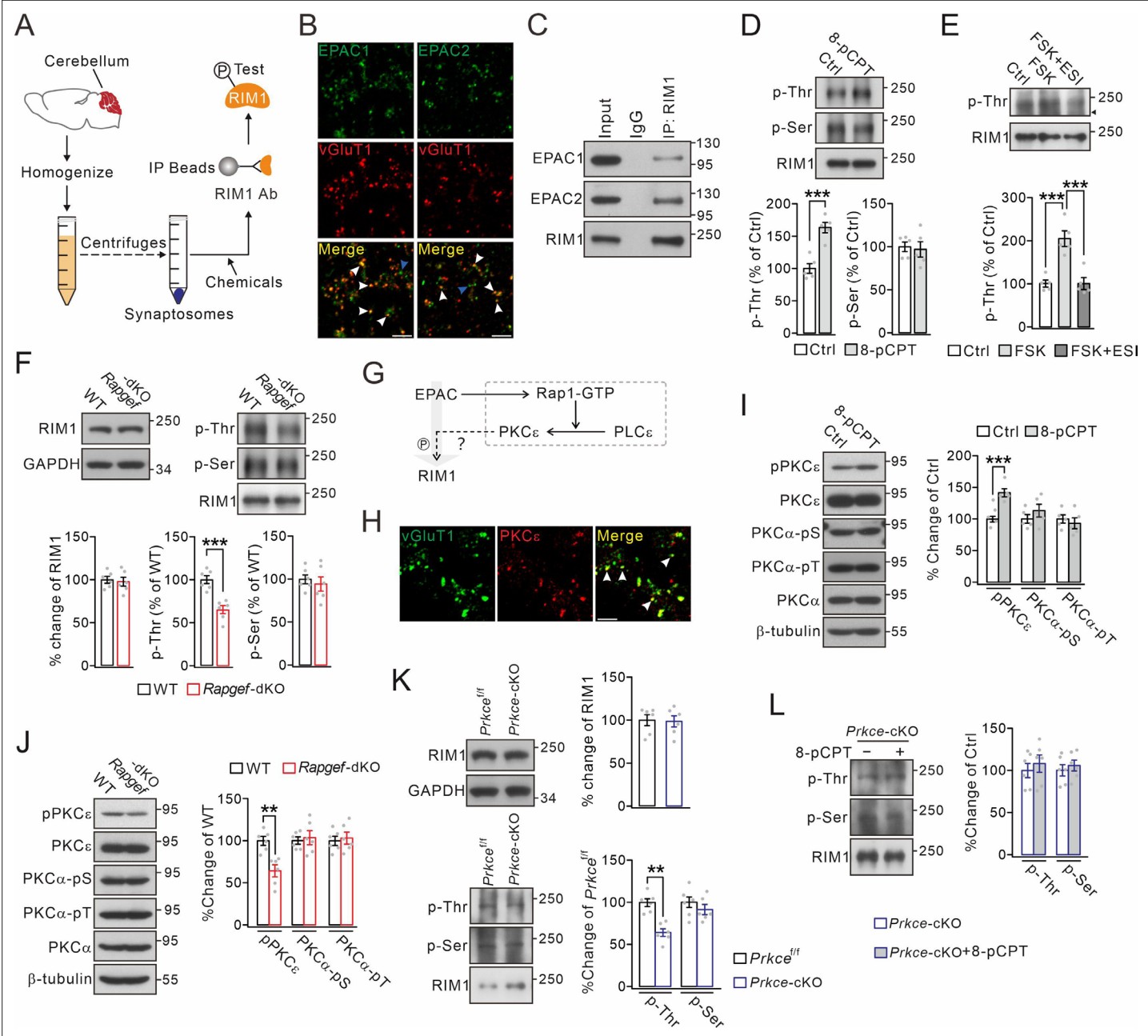

**Figure 1.** Threonine phosphorylation of RIM1 by EPAC and PKCε. (**A**) Schematic showing purification of cerebellar synaptosomes and phophorylation assay of RIM1. (**B**) Immunostaining of EPAC1 or EPAC2 along with vGluT1 (white arrowheads) in cerebellar synaptosomes. Blue arrowheads show the synaptosomes marked by only EPAC1 or EPAC2. Scale bars, 5 μm. (**C**) Precleared synaptosomes (WT) were immunoprecipitated with anti-RIM1 antibody and probed with antibodies to EPAC1, EPAC2 and RIM1. Rabbit IgG was negative control. n=4. (**D**) WT synaptosomes were treated with control buffer (Ctrl) or 8-pCPT (20 μM, 30 min) and p-Thr and p-Ser of RIM1 were analyzed. p-Thr and p-Ser were normalized to corresponding RIM1 and percentage changes relative to Ctrl are plotted. p-Thr: 100 ± 8% (Ctrl) and 163 ± 8% (8-pCPT; p=0.00043). p-Ser: 100 ± 6% (Ctrl) and 97 ± 9% (8-pCPT; p=0.77). Unpaired $t$ test. n=5 for all groups. ***p<0.001. (**E**) p-Thr and p-Ser of RIM1 in WT synaptosomes treated with control buffer, forskolin (FSK; 20 μM, 30 min), or FSK +ESI-09 (50 μM, 30 min) (FSK +ESI). Arrowhead marks nonspecific protein. p-Thr: 100 ± 8% (Ctrl), 205 ± 18% (FSK; p<0.001 *vs.* Ctrl), and 101 ± 14% (FSK +ESI; p=0.98 *vs.* Ctrl; p<0.001 *vs.* FSK). One-way ANOVA test. n=5 for all groups. ***p<0.001. (**F**) Phosphorylation of synaptosomal RIM1 from WT and *Rapgef3/4*-dKO mice. RIM1: 100 ± 4% (WT) and 98 ± 5% (*Rapgef3/4*-dKO; p=0.72). p-Thr: 100 ± 5% (WT) and 65 ± 5% (*Rapgef3/4*-dKO; p=0.00032). p-Ser: 100 ± 5% (WT) and 94 ± 8% (*Rapgef3/4*-dKO; p=0.57). Unpaired $t$ test. n=6 for all groups. ***p<0.001. (**G**) Schematic depiction of proposed working model. The solid lines show known signaling pathways and the dashed line shows the hypothesis. (**H**) Immunostaining of PKCε and vGluT1 (arrowheads) in cerebellar synaptosomes. Scale bar, 5 μm. (**I**) WT synaptosomes were treated with control buffer or 8-pCPT (20 μM, 30 min). The phosphorylations of PKCε and PKCα were normalized to β-tubulin and percentage changes relative to control are plotted. pPKCε: 100 ± 5% (Ctrl)

*Figure 1 continued on next page*

*Figure 1 continued*

and 142 ± 7% (8-pCPT; p=0.0007). PKCα-pSer: 100 ± 8% (Ctrl) and 113 ± 11% (8-pCPT; p=0.31). PKCα-pThr: 100 ± 7% (Ctrl) and 93 ± 10% (8-pCPT; p=0.54). Unpaired *t* test. n=5 for all groups. ***p<0.001. (J) Phosphorylation of synaptosomal PKCε and PKCα in WT and *Rapgef3/4*-dKO mice. pPKCε, PKCα-pSer and PKCα-pThr were normalized to β-tubulin and their percentage changes relative to WT are plotted. pPKCε: 100 ± 5% (WT) and 64 ± 7% (*Rapgef3/4*-dKO; p=0.0013). PKCα-pSer: 100 ± 4% (WT) and 103 ± 8% (*Rapgef3/4*-dKO; p=0.70). PKCα-pThr: 100 ± 6% (WT) and 103 ± 7% (*Rapgef3/4*-dKO; p=0.73). Unpaired *t* test. n=6 for all groups. **p<0.01. (K) Phosphorylation of synaptosomal RIM1 in *Prkce*^f/f and *Prkce*-cKO mice. RIM1: 100 ± 6% (WT) and 99 ± 6% (*Rapgef3/4*-dKO; p=0.88). p-Thr: 100 ± 3% (*Prkce*^f/f) and 65 ± 6% (*Prkce*-cKO; p=0.0028). p-Ser: 100 ± 5% (*Prkce*^f/f) and 95 ± 9% (*Prkce*-cKO; p=0.57). Unpaired *t* test. n=6 for all groups. **p<0.01. (L) Synaptosomes (*Prkce*-cKO) were treated wi/wo 8-pCPT (20 μM, 30 min) and RIM1 phosphorylation was analyzed. p-Thr: 100 ± 8%(*Prkce*-cKO) and 108 ± 10% (*Prkce*-cKO +8 pCPT; p=0.55). p-Ser: 100 ± 7% (*Prkce*-cKO) and 106 ± 6% (*Prkce*-cKO +8 pCPT; p=0.57). Unpaired *t* test. n=6 for all groups. See the online *Figure 1—source data 1* file for source data of western blots in this figure.

The online version of this article includes the following source data and figure supplement(s) for figure 1:

**Source data 1.** The uncut gel of western blots in *Figure 1*.

**Figure supplement 1.** The percentages of PF and CF synapses among cerebellar synaptosomes.

**Figure supplement 2.** Threonine phosphorylation of RIM1 by EPAC in vitro.

**Figure supplement 2—source data 1.** The uncut gel of western blots in *Figure 1—figure supplement 2*.

**Figure supplement 3.** Generation of *Prkce*-cKO and *Rapgef3;Rapgef4*-cKO mice and cerebellar cytology.

**Figure supplement 3—source data 1.** The uncut gel of western blots and PCR in *Figure 1—figure supplement 3*.

**Figure supplement 4.** 8-pCPT-induced RIM1 phosphorylation is blocked by PKCε inhibitor.

**Figure supplement 4—source data 1.** The uncut gel of western blots in *Figure 1—figure supplement 4*.

Since EPAC is an effector of cAMP, we wondered whether cAMP also causes the phosphorylation of threonine sites of RIM1, which comprises 27 of such sites (*Figure 1—figure supplement 2C*). Hence, forskolin, an activator of adenylate cyclase, and ESI-09, an inhibitor of EPAC (*Gutierrez-Castellanos et al., 2017*), were administered to synaptosomes, after which RIM1 p-Thr was measured. RIM1 p-Thr was vastly increased by forskolin alone, but not following co-application of both forskolin and ESI-09 (*Figure 1E*), indicating that cAMP leads to EPAC-dependent threonine phosphorylation of RIM1. We continued to examine the consequences on RIM1 phosphorylation in EPAC1 and EPAC2 double-knockout (*Rapgef3/4*-dKO; *Rapgef3* and *Rapgef4* are genes coding for EPAC1 and EPAC2, respectively) mice. The reason for this strategy is that EPAC1 and EPAC2 share highly conserved cAMP-binding domains, and have significant cross-talk and redundant roles in many physiological processes (*Cheng et al., 2008*). Using synaptosomes purified from *Rapgef3/4*-dKO mice, we found that RIM1 p-Thr was significantly reduced, whereas RIM1 p-Ser was unchanged (*Figure 1F*). Meanwhile, knockout of *Rapgef3/4* did not change the expression of RIM1 (*Figure 1F*). The difference of RIM1 p-Thr in the *Rapgef3/4*-dKO mice was not accompanied by major structural difference, as EPAC deficiency did not interfere with lobule thickness or number of PC spines (*Figure 1—figure supplement 3A and B*). Together, these data strongly indicate that EPAC is necessary and sufficient to induce threonine phosphorylation of RIM1.

EPAC by itself lacks the kinase activity that is required for phosphorylation (*Kawasaki et al., 1998*; *Cheng et al., 2008*), leading to a question how EPAC mediates the phosphorylation of RIM1. We hypothesized that EPAC might act on RIM1 through the Rap1-PLCε-PKCε module (*Figure 1G*), which is shown to be activated by EPAC in neuroblastoma cells (*Schmidt et al., 2001*), dorsal root ganglion neurons (*Hucho et al., 2005*), as well as heart cells (*Oestreich et al., 2009*). Our hypothesis was corroborated by several lines of evidence. First, when Flag-EPAC1 and Flag-EPAC2 were expressed in HEK cells, the phosphorylation at PKCε-S729 was significantly increased by either EPAC1 or EPAC2 expression, whereas the phosphorylation at PKCα-S657 or PKCα-T638 was not altered (*Figure 1—figure supplement 4A*). Second, PKCε overlapped with vGluT1 in cerebellar synaptosomes (*Figure 1H*), suggesting the presence of PKCε at PF synapses. Western blots showed that phosphorylation at PKCε-S729, but not at PKCα-S657 or PKCα-T638, was increased in cerebellar synaptosomes treated with 8-pCPT, whereas control buffer had no impact (*Figure 1I*). Third, phosphorylation at PKCε-S729 in the synaptosomes was significantly reduced by EPAC ablation (*Rapgef3/4*-dKO versus WT), whereas phosphorylation at PKCα-S657 or PKCα-T638 was unchanged (*Figure 1J*). These data indicate that EPAC is able to regulate PKCε activity. We next investigated whether PKCε can phosphorylate RIM1α. HA-RIM1α and His-PKCε were co-transfected into HEK cells and co-IP experiments showed that PKCε

can bind to RIM1α (*Figure 1—figure supplement 4B*). In addition, RIM1α p-Thr was significantly increased in cells transfected with PKCε compared to the control (*Figure 1—figure supplement 4C*). To confirm in vitro findings, we generated mice with *Prkce* (the gene coding for PKCε) deletion specifically in cerebellar granule cells (*Prkce*-cKO) by crossing *Atoh1*[Cre] (*Wang et al., 2020*) with *Prkce*[f/f] mice (*Figure 1—figure supplement 3C–F*), and *Prkce*-cKO mice showed normal lobule thickness and number of spines of PCs (*Figure 1—figure supplement 3G–H*). Subsequently, RIM1 phosphorylation was examined in cerebellar synaptosomes derived from *Prkce*[f/f] and *Prkce*-cKO mice. Similar to the findings in *Rapgef3/4*-dKO mice, RIM1 p-Thr was significantly reduced, whereas both RIM1 p-Ser and total RIM1 were unchanged in *Prkce*-cKO mice (*Figure 1K*). These data indicate that PKCε is able to regulate RIM1 p-Thr phosphorylation.

Finally, several lines of evidence demonstrated the causal relationship between EPAC and PKCε on the phosphorylation of RIM1α. First, we applied 8-pCPT alone or with εV1-2 (a selective PKCε inhibitor) to WT synaptosomes. The addition of εV1-2 to the synaptosomes strongly attenuated RIM1 p-Thr induced by 8-pCPT (*Figure 1—figure supplement 4D*). In contrast, RIM1 p-Thr was not affected by co-application of Gö6976, a PKCα/β inhibitor (*Figure 1—figure supplement 4D*). Second, we administered phorbol 12-myristate 13-acetate (PMA), an activator of all PKC isoforms, alone or along with εV1-2 or Gö6976, so as to inhibit PKCε or PKCα/β, respectively. εV1-2, but not Gö6976, significantly suppressed RIM1α p-Thr in the synaptosomes (*Figure 1—figure supplement 4E*). Third, RIM1 phosphorylation was examined in *Prkce*-cKO synaptosomes, which were treated with either control saline or 8-pCPT. In this scenario, neither p-Thr nor p-Ser of RIM1 was changed (*Figure 1L*). Overall, these data strongly indicate that EPAC can trigger RIM1α p-Thr phosphorylation and that this activation requires PKCε.

## EPAC-PKCε module is critical to vesicle docking and presynaptic release through acting on the Rab3A-RIM1α-Munc13-1 complex

Our finding that the EPAC-PKCε module regulates RIM1 activity through phosphorylation leads to an interesting question: whether the EPAC-PKCε module functions on synaptic formation and function through acting on RIM1, which is known to be critical to organization of the presynaptic active zone and neurotransmitter release (*Schoch et al., 2002*; *Han et al., 2011*; *Kaeser et al., 2011*; *Acuna et al., 2016*; *Persoon et al., 2019*).

To address this question, we first visualized PF-PC synapses using transmission electron microscopy (EM), in which PF boutons were identified by their presence of synaptic vesicles as well as their asymmetric synaptic contacts with PC spines (*Figure 2A and B*). No apparent abnormality was found in the size of the postsynaptic density or the synaptic cleft of PF-PC synapses in either *Rapgef3/4*-dKO (n=4) or *Prkce*-cKO (n=4) mice, compared to corresponding WT (n=4) and *Prkce*[f/f] (n=4) mice (*Figure 2A and B*). However, the deletion of EPAC significantly decreased the number of the docked vesicle pool (WT: 2.0±0.1 vesicles, n=98 boutons; *Rapgef3/4*-dKO: 1.0±0.1 vesicles, n=127 boutons; p<0.0001) (*Figure 2A*). This difference turned out to be specific to the active zone, as the total number of vesicles in PF terminals (within 100 nm away from active zone) was not affected (WT: 32.8±2.4 vesicles, n=98 boutons; *Rapgef3/4*-dKO: 28.3±1.5 vesicles, n=127 boutons; p=0.15). Similarly, the specific deletion of PKCε in granule cells also decreased the number of vesicles in the docked vesicle pool (*Prkce*[f/f]: 1.6±0.1 vesicles, n=60 boutons; *Prkce*-cKO: 0.7±0.1 vesicles, n=66 boutons; p<0.0001) (*Figure 2B*). Meanwhile, the total number of vesicles in PF terminals was also not affected (*Prkce*[f/f]: 42.1±2.3 vesicles, n=57 boutons; *Prkce*-cKO: 39.1±2.6 vesicles, n=55 boutons; p=0.38). These data demonstrate that both EPAC and PKCε regulate the docking of presynaptic vesicles.

We next examined the effect of the ablation of EPAC or PKCε on synaptic transmission. Miniature excitatory synaptic currents (mEPSCs) at PF-PC synapses were recorded in cerebellar slices from *Atoh1*[Cre];*Rapgef3*[f/f];*Rapgef4*[f/f] (*Rapgef3*;*Rapgef4*-cKO) and *Prkce*-cKO mice, the former of which caused specific deletion of *Rapgef3* and *Rapgef4* in granule cells (*Figure 1—figure supplement 3I–L*), while *Atoh1*[Cre] and *Prkce*[f/f] mice were used as corresponding controls. We found that mEPSC frequency was reduced in PCs from *Rapgef3*;*Rapgef4*-cKO mice compared to PCs from *Atoh1*[Cre] mice, whereas mean amplitude did not differ between two genotypes (*Figure 2C*). Similarly, the frequency but not the amplitude of mEPSCs was significantly lower in *Prkce*-cKO mice than corresponding *Prkce*[f/f] mice (*Figure 2D*). A decrease in mEPSC frequency may be due to a reduction in release probability (Pr). To determine if Pr is affected following deletion of presynaptic EPAC and PKCε, we used a

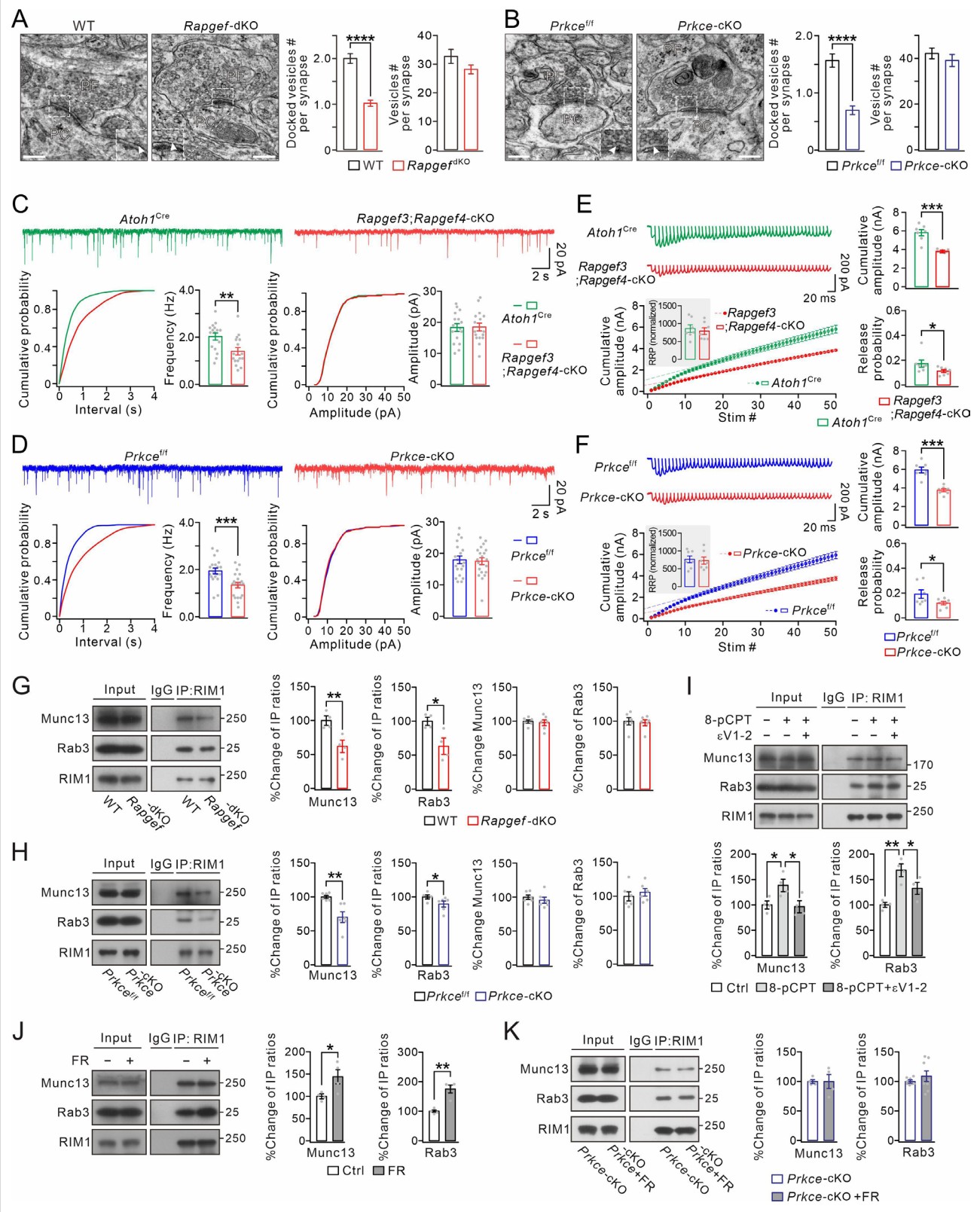

**Figure 2.** EPAC and PKCε act on vesicle docking, synaptic release, and Rab3-RIM1-Munc13 complex. (**A**) Representative EM (23,000×) of PF-PC synapses of WT and *Rapgef3/4*-dKO mice. Scale bars: 200 nm. The inserts show docked vesicles. Unpaired *t* test. ****p<0.0001. (**B**) Representative EM of PF-PC synapses of *Prkce*^f/f^ and *Prkce*-cKO mice. Scale bars: 200 nm. Unpaired *t* test. ****p<0.0001. (**C**) Example PC mEPSCs in *Atoh1*^Cre^ and *Rapgef3;Rapgef4*-cKO mice. Lower: statistics of inter-event interval and amplitude. Grey dots indicate individual data points. Frequency: 2.0±0.2 Hz

*Figure 2 continued on next page*

*Figure 2 continued*

(*Atoh1*[Cre]) and 1.4±0.2 Hz (*Rapgef3;Rapgef4*-cKO; p=0.0036). Amplitude: 18.3±1.3 pA (*Atoh1*[Cre]) and 18.5±1.3 pA (*Rapgef3;Rapgef4*-cKO; p=0.46). Unpaired *t* test. n=for all groups. **p<0.01. (**D**) Example PC mEPSCs from *Prkce*[f/f] and *Prkce*-cKO mice. Frequency: 1.9±0.1 Hz (*Prkce*[f/f]; n=19) and 1.3±0.1 Hz (*Prkce*-cKO; n=20; p=0.00059). Amplitude: 17.9±1.2 pA (*Prkce*[f/f]; n=19) and 17.5±1.1 pA (*Prkce*-cKO; n=20; p=0.39). Unpaired *t* test. ***p<0.001. (**E**) Representative responses of *Atoh1*[Cre] and *Rapgef3;Rapgef4*-cKO PCs to 100 Hz PF stimulation. The artifacts were truncated and each EPSC were aligned to its initial rising point. RRP was defined as the y-intercept of linear portion of cumulative amplitude curve. For RRP (inset), *Atoh1*[Cre]: 861±113; *Rapgef3;Rapgef4*-cKO: 790±101; p=0.31, unpaired *t* test. For cumulative amplitude, *Atoh1*[Cre]: 5815±360 pA; *Rapgef3;Rapgef4*-cKO: 3848±66 pA; p<0.001, unpaired *t* test. For Pr, *Atoh1*[Cre]: 0.17±0.03; *Rapgef3;Rapgef4*-cKO: 0.11±0.01; p=0.043, unpaired *t* test. n=7 for both groups. *p<0.05. ***p<0.001. (**F**) Representative responses of *Prkce*[f/f] and *Prkce*-cKO PCs to 100 Hz PF stimulation. The artifacts were truncated and each EPSC was aligned to its initial rising point. For RRP, *Prkce*[f/f], 764±100; *Prkce*-cKO, 728±106, p=0.40, unpaired *t* test. For cumulative amplitude, *Prkce*[f/f], 5940±337 pA; *Prkce*-cKO, 3755±181 pA; p<0.001, unpaired *t* test. For Pr, *Prkce*[f/f], 0.19±0.04; *Prkce*-cKO, 0.12±0.01; p=0.034, unpaired *t* test. n=7 for both groups. *p<0.05. ***p<0.001. (**G**) Cerebellar synaptosomes from WT and *Rapgef3/4*-dKO mice were immunoprecipitated by anti-RIM1 antibody, and the immunoprecipitates were probed with antibodies to Munc13-1, Rab3A, and RIM1. Rabbit IgG was negative control. Ratios of immunoprecipitated Munc13-1 or Rab3A *vs.* RIM1 were normalized to WT. Munc13-1: 100 ± 6% (WT) and 62 ± 8% (*Rapgef3/4*-dKO; p=0.0081, n=4). Rab3A: 100 ± 5% (WT) and 63 ± 10% (*Rapgef3/4*-dKO; p=0.019, n=4). Total Rab3A and RIM1 were normalized to WT. Munc13-1: 100 ± 2% (WT) and 98 ± 4% (*Rapgef3/4*-dKO; p=0.73, n=6). Rab3A: 100 ± 5% (WT) and 98 ± 4% (*Rapgef3/4*-dKO; p=0.77, n=6). Unpaired *t* test. *p<0.05. **p<0.01. (**H**) Immunoprecipitation of Munc13-1 and Rab3A with RIM1 in cerebellar synaptosomes from *Prkce*[f/f] and *Prkce*-cKO mice. Ratios of immunoprecipitated Munc13-1 or Rab3A *vs.* RIM1 were normalized to WT. Munc13-1: 100 ± 2% (*Prkce*[f/f]) and 70 ± 8% (*Prkce*-cKO; p=0.0030). Rab3A: 100 ± 2% (*Prkce*[f/f]) and 89 ± 4% (*Prkce*-cKO; p=0.019). Total Rab3A and RIM1 were normalized to *Prkce*[f/f]. Munc13-1: 100 ± 3% (*Prkce*[f/f]) and 96 ± 5% (*Prkce*-cKO; p=0.46). Rab3A: 100 ± 7% (*Prkce*[f/f]) and 106 ± 5% (*Prkce*-cKO; p=0.52). n=6 for all groups. Unpaired *t* test. *p<0.05. **p<0.01. (**I**) Cerebellar synaptosomes (WT) mice were incubated in control buffer or 8-pCPT (20 µM, 30 min) and εV1-2 (5 µM, 30 min) and immunoprecipitated. Ratios of immunoprecipitated Munc13-1 or Rab3A *vs.* RIM1 were normalized to control. Munc13-1: 100 ± 8% (Ctrl); 138 ± 12% (8-pCPT; p=0.041 *vs.* Ctrl); 96 ± 12% (8-pCPT+εV1-2; p=0.97 *vs.* Ctrl; p=0.029 *vs* 8-pCPT). Rab3A: 100 ± 5% (Ctrl); 168 ± 12% (8-pCPT; p=0.0011 *vs.* Ctrl); 133 ± 12% (8-pCPT+εV1-2; p=0.069 *vs.* Ctrl; p=0.046 *vs* 8-pCPT). One-way ANOVA test. n=4 for all groups. *p<0.05. **p<0.01. (**J**) Cerebellar synaptosomes (WT) were treated with control buffer or FR236924 (FR) (200 nM, 30 min) and immunoprecipitated. Ratios of immunoprecipitated Munc13-1 or Rab3A *vs.* RIM1 were normalized to Ctrl. Munc13-1: 100 ± 4% (Ctrl) and 144 ± 16% (FR; p=0.041). Rab3A: 100 ± 4% (Ctrl) and 175 ± 13% (FR; p=0.0016). Unpaired *t* test. n=4 for all groups. *p<0.05. **p<0.01. (**K**) Cerebellar synaptosomes (*Prkce*-cKO) were treated with control buffer or FR236924 and immunoprecipitated. Ratios of immunoprecipitated Munc13-1 or Rab3A *vs.* RIM1 were normalized to *Prkce*-cKO. Munc13-1: 100 ± 3% (*Prkce*-cKO; n=4) and 100 ± 12% (*Prkce*-cKO +FR; p=0.99; n=4). Rab3A: 100 ± 2% (*Prkce*-cKO; n=8) and 108 ± 9% (*Prkce*-cKO +FR; p=0.37; n=8). Unpaired *t* test. See the online ***Figure 2—source data 1*** file for source data of western blots in this figure.

The online version of this article includes the following source data and figure supplement(s) for figure 2:

**Source data 1.** The uncut gel of western blots in *Figure 2*.

**Figure supplement 1.** Input-output relationship of evoked PF-EPSCs in control and mutant mice.

repeated stimulation protocol to estimate the readily releasable pool (RRP) as well as Pr (***Thanawala and Regehr, 2016***; ***He et al., 2019***). Compared to *Atoh1*[Cre] and *Prkce*[f/f] mice, repeated stimulation (100 Hz) revealed significant reductions in Pr in *Rapgef3;Rapgef4*-cKO (***Figure 2E***) and *Prkce*-cKO mice (***Figure 2F***). Furthermore, we examined the evoked PF-PC EPSCs with different stimulation intensities (3–15 µA) in control and mutant mice. Our results showed that presynaptic deletion of either EPAC1/ EPAC2 or PKCε significantly decreased evoked EPSCs in response to all stimuli (***Figure 2—figure supplement 1***). These recordings, together with the EM experiment (***Figure 2A and B***), indicate that EPAC-PKCε module is important to presynaptic transmitter release at PF-PC synapses.

We continued to explore how exactly the EPAC-PKCε module modulates synaptic release. An essential process during neurotransmitter release is that Rab3A, RIM1α and Munc13-1 form a tripartite complex and act in concert to dock synaptic vesicles to a release-competent state (***Betz et al., 2001***; ***Wang et al., 2001***; ***Dulubova et al., 2005***). Thus, we investigated whether the EPAC-PKCε module acts on the Rab3A-RIM1α-Munc13-1 complex. By measuring the ratios of IP/input in co-IP assay of synaptosome extracts, we found that both Munc13-1 and Rab3A had significantly weaker binding ability with RIM1α in both *Rapgef3/4*-dKO (***Figure 2G***) and *Prkce*-cKO (***Figure 2H***) synaptosomes, as compared to WT and *Prkce*[f/f] respectively. In contrast, neither EPAC nor PKCε ablation changed the expression levels of Rab3A and Munc13 (***Figure 2G and H***). These data indicate that the deficiency of either EPAC or PKCε impairs protein interactions in the Rab3A-RIM1α-Munc13-1 complex.

In another set of experiments, we studied whether the EPAC-PKCε module is sufficient to boost protein interactions in the Rab3A-RIM1α-Munc13-1 complex. First, we treated WT synaptosomes with 8-pCPT and εV1-2, and measured the amount of Munc13-1 and Rab3A precipitated with RIM1. The quantification showed a significant increment of precipitated Munc13-1 and Rab3A when synaptosomes were incubated with 8-pCPT (***Figure 2I***). Second, we measured the amounts of precipitated Munc13-1 and Rab3A in WT synaptosomes treated with FR236924, a selective activator of PKCε.

We found that precipitations of Munc13-1 and Rab3A were both increased (**Figure 2J**). These data indicate that either EPAC or PKCε is sufficient to promote the formation of the tripartite complex. In parallel experiments, PKCε inhibitor εV1-2 prevented the increase of precipitated Munc13-1 and Rab3A induced by 8-pCPT (**Figure 2I**), while FR236924 failed to induce more precipitations of Munc13-1 and Rab3A in *Prkce*-cKO synaptosomes (**Figure 2K**). In summary, these data demonstrate that the EPAC-PKCε module regulates synaptic organization and transmitter release by regulating the stability of Rab3A-RIM1α-Munc13-1 complex.

## Presynaptic PF-PC LTP depends on EPAC and PKCε

Repetitive stimuli of PF terminals result in an increased Pr of neurotransmitters, leading to the expression of presynaptic LTP (**Salin et al., 1996**; **Kimura et al., 1998**; **van Beugen et al., 2013**; **Hirano et al., 2016**; **Kaeser et al., 2008**; **Yang and Calakos, 2010**; **Martín et al., 2020**). If the EPAC-PKCε module determines transmitter release through regulating the phosphorylation level of RIM1α, it is reasonable to hypothesize that this cascade controls presynaptic PF-PC LTP.

To test this hypothesis, presynaptic LTP at PF-PC synapses was induced by a tetanus stimulation (8 Hz for 5 min) at voltage-clamp mode (–70 mV) (**Figure 3A**). The potentiation of EPSCs reached 131 ± 6% of baseline in WT mice (t=38–40 min; n=13; p<0.001; **Figure 3B and C**), consistent with previous work (**Salin et al., 1996**; **Kaeser et al., 2008**). Concomitantly, paired-pulse facilitation (PPF) ratio decreased to 84 ± 4% (t=38–40 min; n=13; p<0.001; **Figure 3C**), indicating a presynaptic contribution to this form of LTP (**Salin et al., 1996**). Next, we preincubated WT slices with forskolin for 20 min to ensure the effect of forskolin. In this condition, the tetanus stimulation for presynaptic LTP failed to induce synaptic potentiation in PCs (**Figure 3—figure supplement 1A and B**), indicating that presynaptic LTP at PF-PC synapses occurs upon a rise in the cellular level of cAMP.

Next, we examined presynaptic PF-PC LTP in acute slices from *Rapgef3/4*-dKO and *Rapgef3;Rapgef4*-cKO mice. We made whole-cell recordings from PCs and found that 8 Hz stimulation failed to induce potentiation of EPSCs in *Rapgef3/4*-dKO mice (104 ± 5% of baseline at t=38–40 min; n=11; p=0.66) (**Figure 3—figure supplement 1C and D**). This finding was confirmed in recordings from slices of *Rapgef3;Rapgef4*-cKO mice, in which EPAC is deleted from the granule cells innervating the PCs, showing that presynaptic PF-PC LTP was also blocked (93 ± 4% of baseline at t=38–40 min; n=9; p=0.059) (**Figure 3F and G**). In control experiments using *Atoh1*[Cre] mice, the potentiation of EPSCs reached 120 ± 5% of baseline (t=38–40 min; n=10; p=0.004; **Figure 3D and E**). These results indicate that presynaptic EPAC is required for presynaptic LTP.

To determine the role of presynaptic PKCε on presynaptic LTP, we recorded 8 Hz stimulation-induced EPSC potentiation in *Prkce*[f/f] and *Prkce*-cKO mice. Similar to WT and *Atoh1*[Cre] mice, the potentiation of PF-EPSCs evoked by 8 Hz stimulation reached 120 ± 3% of baseline in control *Prkce*[f/f] mice (t=38–40 min; n=7; p=0.004; **Figure 3H and I**). However, presynaptic ablation of PKCε completely blocked the induction of this form of LTP (99 ± 5%; n=10; p=0.065; **Figure 3J and K**), suggesting that presynaptic PF-PC LTP also requires PKCε. Here too, the PPF ratio was unaffected (p=0.77 at t=38–40 min; n=10; **Figure 3K**). This conclusion was further confirmed following chemical inhibition of PKCε by continuously administering εV1-2 to cerebellar slices from WT mice, as εV1-2 completely blocked the induction of presynaptic PF-PC LTP (101 ± 4%; n=9; p=0.59; **Figure 3—figure supplement 1E and F**).

Cerebellar synaptic plasticity might be affected by animal age, recording temperature, and Ca²⁺ concentration in aCSF. To better illustrate the role of EPAC and PKCε in presynaptic PF-LTP, we changed experimental conditions and revisited presynaptic PF-LTP. First, we examined presynaptic PF-PC LTP in 2-month-old *Rapgef3/4*-dKO and *Rapgef3;Rapgef4*-cKO mice. In control experiments with *Atoh1*[Cre] mice, 8 Hz stimulation induced a potentiation of EPSCs (120 ± 4% of baseline at t=38–40 min; n=7; p=0.002) (**Figure 3—figure supplement 2A**). In contrast, the same stimulation failed to induce the potentiation in *Rapgef3;Rapgef4*-cKO mice (97 ± 1% of baseline at t=38–40 min; n=6; p=0.76) (**Figure 3—figure supplement 2B**). Similarly, presynaptic PF-PC LTP was successfully induced in 2-month-old *Prkce*[f/f] mice (119 ± 1% of baseline at t=38–40 min; n=6; p<0.001) (**Figure 3—figure supplement 2C**), but not in *Prkce*-cKO mice of the same age (98 ± 2% of baseline at t=38–40 min; n=7; p=0.34) (**Figure 3—figure supplement 2D**).

Second, presynaptic PF-PC LTP was recorded at elevated temperature (32 °C) in cerebellar slices from *Atoh1*[Cre], *Rapgef3;Rapgef4*-cKO, *Prkce*[f/f], and *Prkce*-cKO mice (P21). We found that presynaptic LTP was induced in *Atoh1*[Cre] (124 ± 2% of baseline at t=38–40 min; n=6; p=0.00012) and *Prkce*[f/f] mice

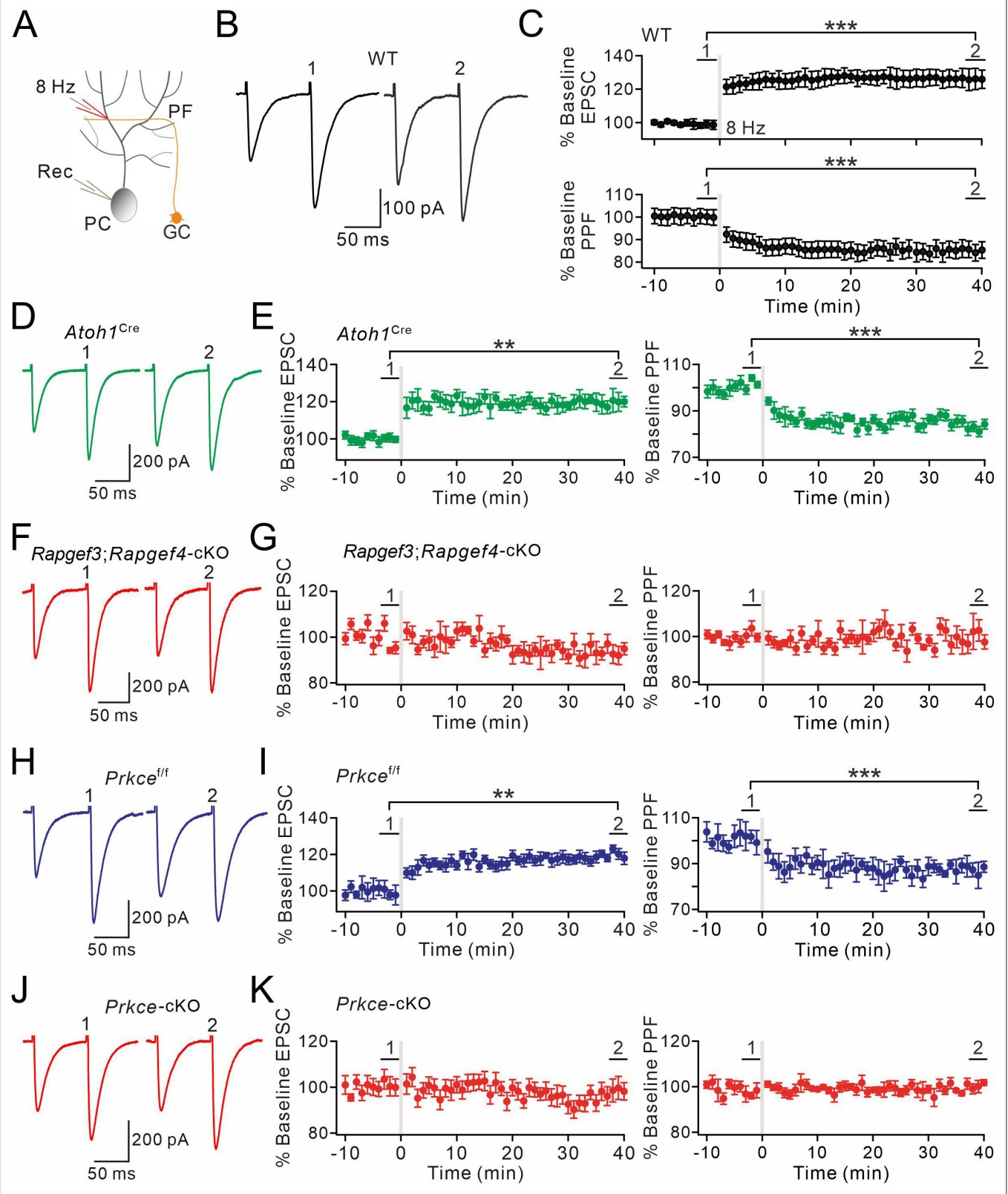

**Figure 3.** EPAC and PKCε are required for presynaptic PF-PC LTP. (**A**) Schematic showing the induction of presynaptic LTP. (**B, D, F, H, J**) Example PF-EPSCs for baseline (1) and after LTP induction (2) in WT (**B**), *Atoh1*^Cre (**D**), *Rapgef3;Rapgef4*-cKO (**F**), *Prkce*^f/f (**H**), and *Prkce*-cKO (**J**) mice. (**C**) Percentage changes of PF-EPSC amplitudes (WT). (1): 101 ± 4%; (2): 131 ± 6%; n=13; p<0.001. Percentage changes of PPF ratios from cells shown above. (1): 101 ± 3%; (2): 84 ± 4%; n=13; p<0.001. Paired *t* test. ***p<0.001. (**E**) Left: percentage changes of PF-EPSC amplitudes (*Atoh1*^Cre). (1): 100 ± 2%; (2): 120 ±

*Figure 3 continued on next page*

*Figure 3 continued*

5%; n=10; p=0.004. Right: percentage changes of PPF ratios. (1): 102 ± 2%; (2): 83 ± 2%; n=10; p<0.001. Unpaired *t* test. **p<0.01. ***p<0.001. (**G**) Left: percentage changes of PF-EPSC amplitudes (*Rapgef3;Rapgef4*-cKO). (1): 99 ± 2%; (2): 93 ± 4%; n=9; p=0.059. Right: percentage changes of PPF ratios. (1): 101 ± 3%; (2): 101 ± 5%; n=9; p=0.07. Paired *t* test. (**I**) Left: percentage changes of PF-EPSC amplitudes (*Prkce*[f/f]). (1): 99 ± 4%; (2): 120 ± 3%; n=7; p=0.004. Right: percentage changes of PPF ratios. (1): 100 ± 5%; (2): 86 ± 4%; n=7; p<0.001. Paired *t* test. **p<0.01. ***p<0.001. (**K**) Left: percentage changes of PF-EPSC amplitudes (*Prkce*-cKO). (1): 101 ± 4%; (2): 99 ± 5%; n=10; p=0.065. Right: percentage changes of PPF ratios. (1): 101 ± 3%; (2): 100 ± 2%; n=10; p=0.77. Paired *t* test.

The online version of this article includes the following figure supplement(s) for figure 3:

**Figure supplement 1.** Presynaptic PF-PC LTP is blocked by forskolin incubation, EPAC ablation, or εV1-2 application.

**Figure supplement 2.** The induction of presynaptic PF-PC LTP in 2-month-old mice.

**Figure supplement 3.** The induction of presynaptic PF-PC LTP in elevated temperature.

**Figure supplement 4.** The induction of presynaptic PF-PC LTP in lower $Ca^{2+}$ concentration.

(122 ± 1% of baseline at t=38–40 min; n=6; p<0.001) (*Figure 3—figure supplement 3A and C*), but not in *Rapgef3;Rapgef4*-cKO (97 ± 3% of baseline at t=38–40 min; n=6; p=0.33) and *Prkce*-cKO (100 ± 1% of baseline at t=38–40 min; n=6; p=0.92) mice (*Figure 3—figure supplement 3B and D*).

Third, we lowered the $Ca^{2+}$ concentration in the aCSF from 2 mM to 0.5 mM, and then recorded presynaptic LTP in cerebellar slices from *Atoh1*[Cre], *Rapgef3;Rapgef4*-cKO, *Prkce*[f/f], and *Prkce*-cKO mice (P21). Again, presynaptic LTP was induced in *Atoh1*[Cre] (113 ± 2% of baseline at t=38–40 min; n=6; p=0.002) and *Prkce*[f/f] mice (113 ± 5% of baseline at t=38–40 min; n=6; p=0.002) (*Figure 3—figure supplement 4A and C*), but not in *Rapgef3;Rapgef4*-cKO (99 ± 2% of baseline at t=38–40 min; n=6; p=0.61) and *Prkce*-cKO (104 ± 2% of baseline at t=38–40 min; n=6; p=0.53) mice (*Figure 3—figure supplement 4B and D* ). These experiments indicate that the function of EPAC and PKCε in the induction of presynaptic PF-LTP is independent of animal age, recording temperature, or external $Ca^{2+}$ concentration.

Therefore, on the basis of our experiments in PCs from mice with presynaptic specific deletion of EPAC and PKCε, we conclude that the presynaptic EPAC-PKCε module is critical for presynaptic PF-PC LTP in the cerebellum.

## EPAC and PKCε mediate cAMP-triggered EPSC potentiation

cAMP is also required for presynaptic LTP induced by electrical stimulation (*Salin et al., 1996*; *Le Guen and De Zeeuw, 2010*), and its agonists are enough to produce a prominent increase in glutamate release (*Weisskopf et al., 1994*; *Salin et al., 1996*). Next, we wondered which downstream effector, EPAC or PKA (*Cheng et al., 2008*), is responsible for cAMP-induced potentiation. The role of PKA in presynaptic LTP has been contradicted by the studies showing that presynaptic LTP is intact when serine phosphorylation of RIM1 by PKA is interrupted (*Kaeser et al., 2008*; *Yang and Calakos, 2010*; also see *Lonart et al., 2003*). Moreover, *Martín et al., 2020* showed that EPAC2 regulates synaptic release at PF synapses and is required for presynaptic PF-PC LTP. These findings inspired us to investigate whether perhaps the EPAC-PKCε module mediates cAMP-triggered EPSC potentiation.

We made whole-cell recordings from PCs and evoked PF-EPSCs every 30 s in *Atoh1*[Cre], *Rapgef3;Rapgef4*-cKO and *Prkce*-cKO mice. In *Atoh1*[Cre] control mice, external application of forskolin produced a long-lasting elevation in PF-EPSC amplitude (*Figure 4A and B*), with a peak potentiation of 366 ± 25% (at 48–50 min; n=15; *Figure 4C*). In contrast, simultaneous ablation of EPAC1 and EPAC2 at presynaptic sites prominently affected the synaptic potentiation induced by forskolin application (162 ± 18% at 48–50 min; n=12; *Figure 4A–C*). Next, we incubated *Rapgef3;Rapgef4*-cKO PCs along with PKA antagonist KT5720 (3 μM) and again examined forskolin-induced EPSC potentiation. In this case, we found that combined blockade of EPAC and PKA completely eliminated the action of forskolin on EPSC potentiation (106 ± 4% at 48–50 min; n=12; *Figure 4A–C*). We continued to examine the effect of PKCε on cAMP-triggered EPSC potentiation using *Prkce*-cKO mice. Similar to *Rapgef3;Rapgef4*-cKO mice, the forskolin-induced potentiation in *Prkce*-cKO PCs was significantly attenuated (198 ± 5% at 48–50 min; n=12; *Figure 4A–C*). Again, the remaining potentiation was further blocked by the addition of KT5720 (101 ± 3% at 48–50 min; n=12; *Figure 4A–C*). The inhibitory effect of KT5720 on forskolin-induced potentiation was also examined by applying it alone in *Atoh1*[Cre] PCs. We found that KT5720 inhibited the potentiation by 15%, a smaller effect than that

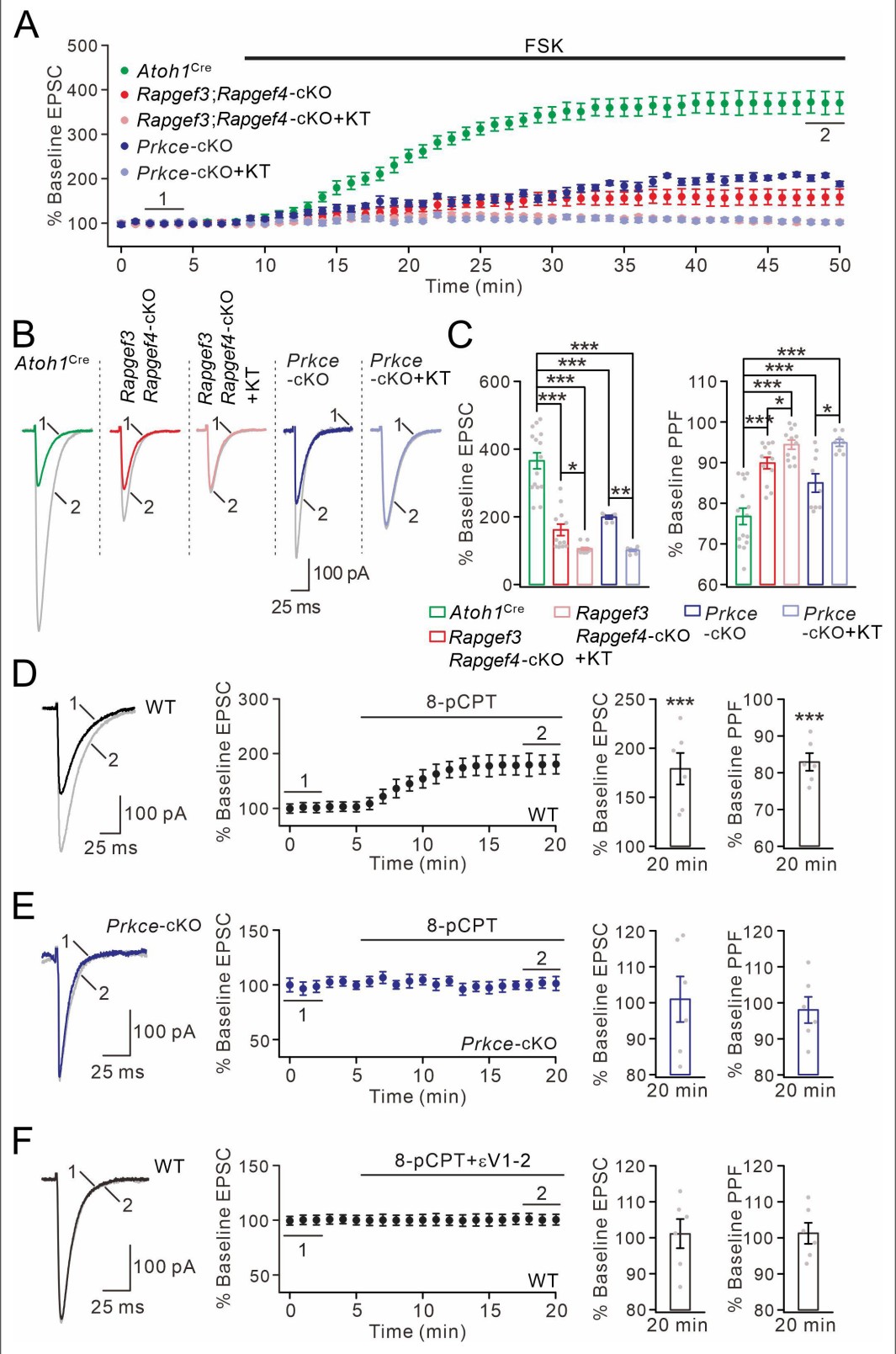

**Figure 4.** cAMP-triggered PF facilitation is dependent on EPAC and PKCε. (**A**) The facilitation of PF-EPSCs by forskolin (FSK) (20 μM) in *Atoh1*Cre, *Rapgef3;Rapgef4*-cKO and *Prkce*-cKO mice. (**B**) Example traces for baseline (1) and after potentiation (2) shown in (**A**). (**C**) Left: percent changes of EPSC amplitude. *Atoh1*Cre: 366 ± 25% (n=15); *Rapgef3;Rapgef4*-cKO: 162 ± 18% (n=12; p<0.001 *vs. Atoh1*Cre); *Rapgef3;Rapgef4*-cKO+KT: 106 ± 4% (n=12;

*Figure 4 continued on next page*

*Figure 4 continued*

p<0.001 *vs. Atoh1*^Cre; p=0.046 *vs. Rapgef3;Rapgef4*-cKO); *Prkce*-cKO: 198 ± 5% (n=12; p<0.001 *vs. Atoh1*^Cre); *Prkce*-cKO +KT: 101 ± 3% (n=12; p<0.001 *vs. Atoh1*^Cre; p=0.0034 *vs. Prkce*-cKO). Right: percent changes of PPF. *Atoh1*^Cre: 77 ± 2% (n=15); *Rapgef3;Rapgef4*-cKO: 90 ± 1% (n=12; p<0.001 *vs. Atoh1*^Cre); *Rapgef3;Rapgef4*-cKO+KT: 94 ± 1% (n=12; p<0.001 *vs. Atoh1*^Cre; p=0.049 *vs. Rapgef3;Rapgef4*-cKO); *Prkce*-cKO: 85 ± 2% (n=12; p<0.001 *vs. Atoh1*^Cre); *Prkce*-cKO +KT: 95 ± 1% (n=12; p<0.001 *vs. Atoh1*^Cre; p=0.025 *vs. Prkce*-cKO). One-way ANOVA test. *p<0.05. ***p<0.001. (**D**) Bath application of 8-pCPT (20 µM) caused PF-EPSC potentiation in WT mice. Left: example traces before (1) and after potentiation (2). Middle: time course of PF facilitation. Right: percent changes of EPSC amplitude (179 ± 18%; n=6; p<0.001) and PPF (83 ± 3%; n=6; p<0.001) at 18–20 min *vs.* baseline (0–2 min). Paired *t* test. ***p<0.001. (**E**) 8-pCPT failed to induce PF-EPSC potentiation in *Prkce*-cKO mice. Left: example traces for baseline (1) and after potentiation (2). Middle: time course of PF facilitation. Right: percent changes of EPSC amplitude (101 ± 6%; n=6; p=0.35) and PPF (98 ± 4%; n=6; p=0.45) at 18–20 min *vs.* baseline (0–2 min). Paired *t* test. (**F**) Co-application of 8-pCPT and εV1-2 (5 µM) failed to produce PF potentiation in WT mice. Left: example traces for baseline (1) and after potentiation (2). Middle: time course of PF-EPSCs. Right: percent changes of EPSC amplitude (101 ± 4%; n=6; p=0.78) and PPF (101 ± 3%; n=6; p=0.67) at 18–20 min *vs.* baseline (0–2 min). Paired *t* test.

The online version of this article includes the following figure supplement(s) for figure 4:

**Figure supplement 1.** PKA inhibition has a modest effect in blocking cAMP-triggered facilitation.

of EPAC or PKCε ablation (*Figure 4—figure supplement 1*). Thus, these results indicate that EPAC, PKCε and PKA all mediate cAMP-induced potentiation of transmitter release. In parallel with the observation of EPSC amplitude, PPF was monitored during the whole cell recordings. Forskolin application led to a significant reduction in PPF ratio of PF-EPSCs in *Atoh1*^Cre mice (*Figure 4C*). However, this reduction was significantly less when presynaptic of both types of EPAC as well as PKCε were ablated and KT5720 was added (*Figure 4C*). These results highlight that EPAC and PKCε function synergically on the synaptic release at PF-PC synapses.

We next assessed the impact of the EPAC-PKCε module on the strength of PF-EPSCs by directly applying EPAC agonist 8-pCPT. In line with previous work (*Kaneko and Takahashi, 2004*; *Gekel and Neher, 2008*), the administration of 8-pCPT was sufficient to potentiate PF-EPSCs by 179 ± 18% and reduce their PPF ratio by 17 ± 3% in WT PCs (n=6; at 18–20 min) (*Figure 4D*). Two lines of evidence confirm that the potentiation of PF-EPSCs by EPAC is mediated by PKCε. First, 8-pCPT-induced potentiation of PF-EPSCs was diminished in *Prkce*-cKO mice, as shown by unchanged PF-EPSCs and PPF (*Figure 4E*). Second, co-application of εV1-2 effectively prevented the 8-pCPT-induced synaptic potentiation and change in PPF (*Figure 4F*).

In summary, we conclude that EPAC-PKCε module and PKA are both downstream effectors of cAMP, but the EPAC-PKCε module plays the most prominent role in cAMP-triggered EPSC potentiation.

## Presynaptic EPAC and PKCε are not involved in postsynaptic forms of plasticity

The mechanisms for postsynaptic LTP and LTD at PF-PC synapses can be complicated, in that they may depend not only on postsynaptic processes, but sometimes also on presynaptic events (*Le Guen and De Zeeuw, 2010*; *Wang et al., 2014*; *Schonewille et al., 2021*). For example, an endocannabinoid-triggered reduction of synaptic release is required by the induction of postsynaptic LTD (*Kreitzer et al., 2002*). As both EPAC and PKCε regulate Pr of PF-PC synapses, we wondered whether the EPAC-PKCε module also regulates postsynaptic LTP and LTD.

After acquiring stable EPSCs in voltage-clamp mode (–70 mV), we induced postsynaptic LTP by stimulating PFs at 1 Hz for 5 min in current-clamp mode (*Figure 5A*). In WT mice, this tetanus stimulation induced an increase of PF-EPSCs (131 ± 5% of baseline at t=38–40 min; n=13; p<0.001) (*Figure 5B and C*), while PPF was not changed (*Figure 5D*). When this protocol was applied at PF-PC synapses in *Rapgef3/4*-dKO mice, we did not find any sign of potentiation of PF-EPSCs (106 ± 6% of baseline at t=38–40 min; n=13; p=0.26) (*Figure 5B–D*). While these results were consistent with our previous observation that EPAC is required for postsynaptic LTP (*Gutierrez-Castellanos et al., 2017*), we had yet to specify the cellular site of action for EPAC. Therefore, we repeated the induction protocol for postsynaptic LTP in *Atoh1*^Cre and *Rapgef3;Rapgef4*-cKO mice. In this case, the protocol successfully induced PF-PC LTP in both types of mice (*Figure 5E and F*), while PPF was not altered

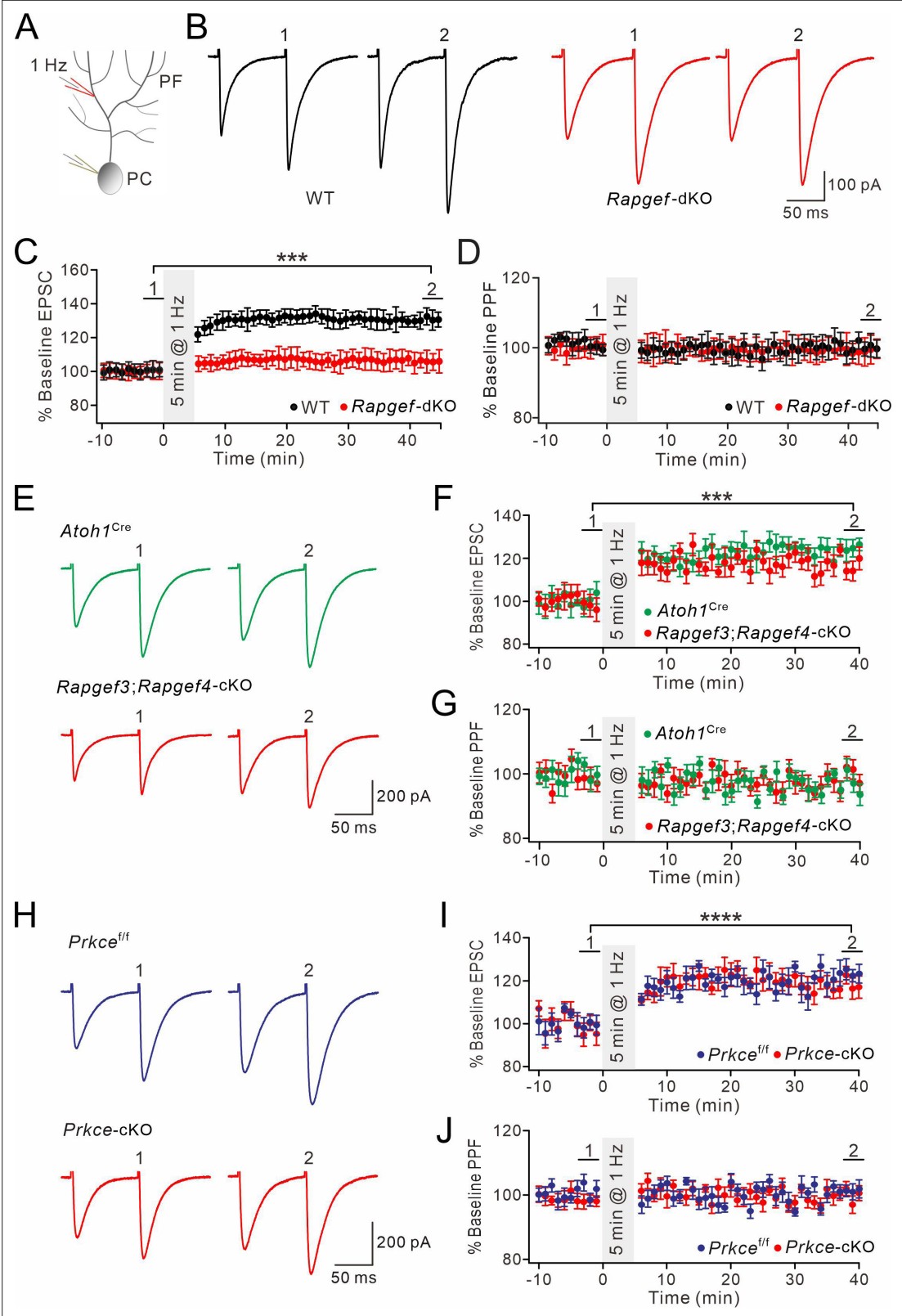

**Figure 5.** Postsynaptic PF-PC LTP is intact upon presynaptic deletion of EPAC or PKCε. (**A**) Schematic showing the induction of postsynaptic LTP. (**B, E, H**) Example PF-EPSCs for baseline (1) and after induction (2) in WT and *Rapgef3/4*-dKO PCs (**B**), *Atoh1*^Cre and *Rapgef3;Rapgef4*-cKO PCs (**E**), and *Prkce*^f/f and *Prkce*-cKO PCs (**H**). (**C**) Percentage changes of PF-EPSC amplitude. In WT, 101 ± 5% for (1) and 131 ± 5% for (2) (p<0.001). In *Rapgef3/4*-dKO, 100 ± 5% for (1) and 106 ± 6% for (2) (p=0.26). Paired *t* test. n=13 for both groups. ***p<0.001. (**D**) Percentage changes of PPF ratios of cells shown in (**C**). In

*Figure 5 continued on next page*

*Figure 5 continued*

WT, 100 ± 2% for (1) and 100 ± 3% for (2) (p=0.63). In *Rapgef3/4*-dKO, 101 ± 3% for (1) and 99 ± 4% for (2) (p=0.74). Paired *t* test. n=13 for both groups. (**F**) Percentage changes of PF-EPSC amplitude. In *Atoh1*[Cre], 100 ± 5% for (1) and 123 ± 3% for (2) (p<0.001). In *Rapgef3;Rapgef4*-cKO, 98 ± 5% for (1) and 119 ± 4% for (2) (p<0.001). Paired *t* test. n=7 for both groups. ***p<0.001. (**G**) Percentage changes of PPF ratios of cells shown in (**C**). In *Atoh1*[Cre]: 100 ± 2% for (1) and 96 ± 3% for (2) (p=0.26). In *Rapgef3;Rapgef4*-cKO: 98 ± 3% for (1) and 95 ± 3% for (2) (p=0.28). Paired *t* test. n=7 for both groups. (**I**) Percentage changes of PF-EPSC amplitude. In *Prkce*[f/f], 99 ± 4% for (1) and 121 ± 4% for (2) (p<0.0001). In *Prkce*-cKO: 97 ± 5% for (1) and 118 ± 5% for (2) (p<0.0001). Paired *t* test. n=7 for both groups. ****p<0.0001. (**J**) Percentage changes of PPF ratios from cells shown in (**I**). In *Prkce*[f/f], 102 ± 2% for (1) and 101 ± 2% for (2) (p=0.73). In *Prkce*-cKO, 98 ± 2% and 100 ± 2% for (2) (p=0.78). Paired *t* test. n=7 for both groups.

(*Figure 5G*), suggesting that this is a postsynaptic form of LTP. We continued to examine the expression of postsynaptic PF-PC LTP in *Prkce*[f/f] and *Prkce*-cKO mice. Similar to *Rapgef3;Rapgef4*-cKO mice, *Prkce*[f/f] and *Prkce*-cKO PCs exhibited robust PF-PC LTP when 1 Hz stimulation was delivered to PFs (*Figure 5H and I*) with unaltered PPF (*Figure 5J*), confirming the postsynaptic site of LTP.

Next, we investigated whether the expression of postsynaptic PF-PC LTD is affected by ablation of EPAC and PKCε. PF-PC LTD was induced by giving repetitive PF stimulation at 100 Hz for 100ms paired with a depolarization of the PCs involved (*Figure 6A*; *Steinberg et al., 2006*; *Zhou et al., 2015*). As shown by example responses (*Figure 6B*), *Rapgef3/4*-dKO PCs showed robust PF-PC LTD (t=38–40 min: 59 ± 4% of baseline; n=13; *Figure 6C*), while the PPF ratio was not changed (p=0.26 at t=38–40 min; n=13; *Figure 6D*). Likewise, PF-PC LTD could be successfully induced in *Atoh1*[Cre] and *Rapgef3;Rapgef4*-cKO mice (*Figure 6E and F*), while PPF was not altered (*Figure 6G*). Moreover, we found that the same protocol could induce PF-PC LTD in *Prkce*[f/f] and *Prkce*-cKO mice (*Figure 6H and I*) without affecting PPF (*Figure 6J*).

Overall, our results suggest that presynaptic EPAC and PKCε are not required for the induction of postsynaptic forms of LTP and LTD.

## The EPAC-PKCε module is essential for motor performance and motor learning

Even though plastic changes in the granular layer of the cerebellum have been suggested to contribute to procedural memory formation (*Le Guen and De Zeeuw, 2010*), the evidence thus far is limited (*Andreescu et al., 2011*; *Galliano et al., 2013*). Therefore, we investigated whether the EPAC-PKCε module, which is critical to presynaptic PF-PC LTP, contributes to performance and adaptation of compensatory eye movements mediated by the vestibulo-cerebellum (*Schonewille et al., 2010*; *Grasselli et al., 2020*).

Basic performance parameters included amplitude (gain) and timing (phase) of the optokinetic response (OKR), vestibulo-ocular reflex (VOR), and visually enhanced VOR (VVOR) (*Figure 7A*). We found that basic motor performance was impaired in *Rapgef3/4*-dKO mice in that they showed significant deficits in the amplitude and timing of their OKR (p=0.009 and p=0.004, respectively; ANOVA for repeated measurements) and VOR (p=0.001 and p=0.02, respectively; ANOVA for repeated measurements) (*Figure 7—figure supplement 1A and B*). In contrast, no significant differences were observed in the VVOR (p=0.66 and p=0.68 for gain and phase values, respectively; *Figure 7—figure supplement 1C*).

The same compensatory eye movements were also tested in *Rapgef3;Rapgef4*-cKO and *Prkce*-cKO mice as well as their littermate controls. Basic eye movement performance was also affected in *Rapgef3;Rapgef4*-cKO mice in that their OKR gains were significantly lower than those of *Atoh1*[Cre] littermates (p=0.003; ANOVA for repeated measurements) (*Figure 7B*), that their VOR gains were significantly greater than those of *Atoh1*[Cre] littermates (VOR: p=0.027; ANOVA for repeated measurements) (*Figure 7C*), and that the phase values during both OKR and VOR were significantly lagging those of the *Atoh1*[Cre] littermates (OKR: p=0.001; VOR: p=0.047; ANOVA for repeated measurements) (*Figure 7B and C*). No significant differences were observed between *Rapgef3;Rapgef4*-cKO and *Atoh1*[Cre] mice in the VVOR (p=0.69 and p=0.75 for gain and phase values, respectively) (*Figure 7D*). Moreover, *Prkce*-cKO mice shared the same defects with *Rapgef3;Rapgef4*-cKO mice in their basic motor performance. OKR gain values of *Prkce*-cKO mice were significantly lower than those of *Prkce*[f/f] littermates (p=0.013; ANOVA for repeated measurements) (*Figure 7E*), whereas their VOR gain values were greater than those of control littermates (p=0.034; ANOVA for repeated measurements) (*Figure 7F*). Meanwhile, OKR and VOR phase values of *Prkce*-cKO mice were both significantly lagging

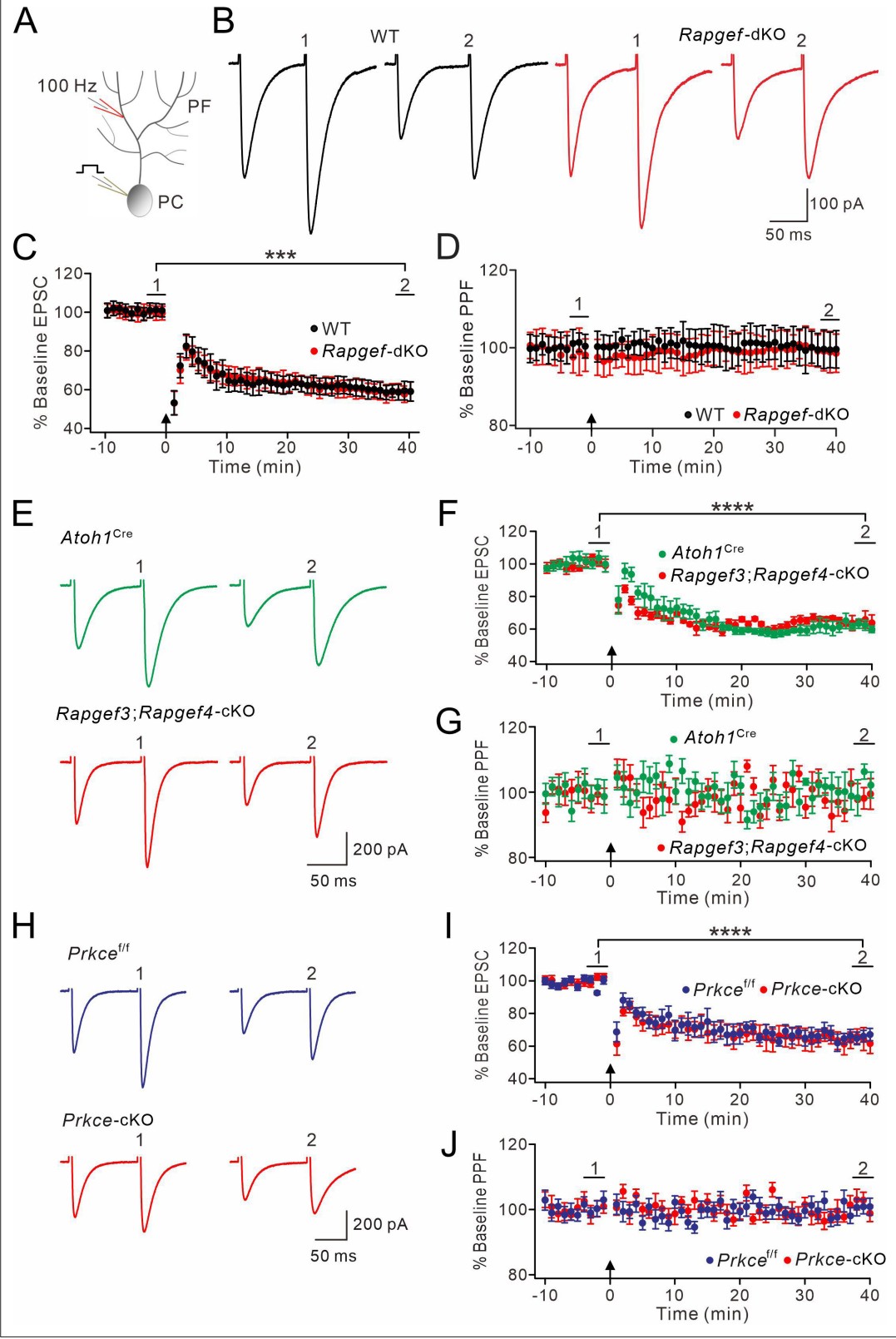

**Figure 6.** PF-LTD is unaltered by presynaptic deletion of EPAC or PKCε. (**A**) A scheme showing the induction of postsynaptic LTD. (**B, E, H**) Example PF-EPSCs for baseline (1) and after LTD induction (2) in WT and *Rapgef3/4*-dKO PCs (**B**), *Atoh1*^Cre and *Rapgef3;Rapgef4*-cKO PCs (**E**), and *Prkce*^f/f and *Prkce*-cKO PCs (**H**). (**C**) Percentage changes of PF-EPSC amplitude. In WT, 101 ± 3% for (1) and 59 ± 5% for (2) (p<0.001). In *Rapgef3/4*-dKO, 100 ± 3%

*Figure 6 continued on next page*

*Figure 6 continued*

for (1) and 59 ± 4% for (2) (<0.001). Paired *t* test. n=13 for both groups. \*\*\*p<0.001. (**D**) Percentage changes of PPF ratios of cells shown in (**C**). In WT, 100 ± 3% for (1) and 100 ± 5% for (2) (p=0.49). In *Rapgef3/4*-dKO, 100 ± 5% for (1) and 100 ± 5% for (2) (p=0.26). Paired *t* test. n=13 for both groups. (**F**) Percentage changes of PF-EPSC amplitude. In *Atoh1*Cre, 100 ± 4% for (1) and 61 ± 3% for (2) (p<0.0001). In *Rapgef3;Rapgef4*-cKO, 101 ± 3% for (1) and 65 ± 4% for (2) (p<0.0001). Paired *t* test. n=7 for both groups. \*\*\*\*p<0.0001. (**G**) Percentage changes of PPF ratios of cells shown in (**F**). In *Atoh1*Cre, 100 ± 2% for (2) and 100 ± 3% for (2) (p=0.40). In *Rapgef3;Rapgef4*-cKO, 101 ± 3% for (2) and 99 ± 4% for (2) (p=0.61). Paired *t* test. n=7 for both groups. (**I**) Percentage changes of PF-EPSC amplitude. In *Prkce*f/f, 99 ± 2% for (1) and 66 ± 4% for (2) (p<0.0001). In *Prkce*-cKO, baseline: 101 ± 2% for (1) and 64 ± 6% for (2) (p<0.0001). Paired *t* test. n=7 for both groups. \*\*\*\*p<0.0001. (**J**) Percentage changes of PPF ratios of cells shown in (**I**). In *Prkce*f/f, 101 ± 2% for (1) and 101 ± 2% for (2) (p=0.56). In *Prkce*-cKO, 100 ± 2% for (1) and 102 ± 2% for (2) (p=0.54). Paired *t* test. n=7 for both groups.

those of the *Atoh1*Cre littermates (OKR: p=0.015; VOR: p=0.044; ANOVA for repeated measurements) (*Figure 7E and F*). No significant differences were observed between *Prkce*f/f and *Prkce*-cKO mice in the VVOR (p=0.93 and p=0.50 for gain and phase values, respectively) (*Figure 7G*). Altogether, our data suggest that presynaptic ablation of EPAC and/or PKCε mice profoundly influences motor performance when visual and vestibular systems are separated, but not when they are engaged simultaneously, as occurs under natural conditions or during visuo-vestibular training.

Next, we tested the VOR phase-reversal protocol, which is considered the type of motor learning sensitive to the perturbation to the vestibulo-cerebellum (*Wulff et al., 2009*; *Badura et al., 2016*; *Peter et al., 2016*). VOR phase reversal aims to reverse the direction of the VOR using retinal slip caused by a screen rotation in the same direction (i.e. in phase) as head rotation and with increasing amplitude as the training progresses (*Figure 7H*). During the initial days of gain-decrease training, all three control mouse lines (WT, *Atoh1*Cre and *Prkce*f/f) exhibited gain reductions similar to previous work (*Wulff et al., 2009*; *Badura et al., 2016*; *Gutierrez-Castellanos et al., 2017*). Gain reductions were smaller in *Rapgef3/4*-dKO (*Figure 7—figure supplement 1D*), *Rapgef3;Rapgef4*-cKO (*Figure 7I*), as well as *Prkce*-cKO (*Figure 7J*) mice, but the deficit varied across days between the different mouse lines (in *Rapgef3/4*-dKO mice, Day 1: p=0.043; Day 2: p=0.008; Day 3: p=0.002; Day 4: p=0.007; Day 5: p=0.004; in *Rapgef3;Rapgef4*-cKO mice, Day 1: p=0.079; Day 2: p=0.036; Day 3: p=0.011; Day 4: p=0.22; Day 5: p=0.061; and in *Prkce*-cKO mice, Day 1: p=0.047; Day 2: p=0.004; Day 3: p=0.004; Day 4: p=0.084; Day 5: p=0.15). WT (*Figure 7—figure supplement 1D*), *Atoh1*Cre (*Figure 7I*) as well as *Prkce*f/f (*Figure 7J*) mice showed a proper reversal of the phase of their VOR, highlighting their ability to invert the direction of an innate reflex (*Wulff et al., 2009*; *Badura et al., 2016*; *Peter et al., 2016*; *Grasselli et al., 2020*). Whereas the VOR phase values were not significantly affected in the *Rapgef3/4*-dKO, *Rapgef3;Rapgef4*-cKO, and *Prkce*-cKO mouse lines during the first day (WT versus *Rapgef3/4*-dKO, p=0.15; *Atoh1*Cre versus *Rapgef3;Rapgef4*-cKO, p=0.087; *Prkce*f/f versus *Prkce*-cKO, p=0.52), they were so during sessions on days 2–5 (WT versus *Rapgef3/4*-dKO: Day 2, p=0.003; Day 3, p=0.002; Day 4, p<0.001; Day 5, p<0.001; *Atoh1*Cre versus *Rapgef3;Rapgef4*-cKO: Day 2, p<0.001; Day 3, p<0.001; Day 4, p<0.001; Day 5, p<0.001; *Prkce*f/f versus *Prkce*-cKO: Day 2, p=0.01; Day 3, p=0.048; Day 4, p<0.001; Day 5, p<0.001). Therefore, we conclude that *Rapgef3/4*-dKO, *Rapgef3;Rapgef4*-cKO and *Prkce*-cKO mice had prominent deficits in phase-reversal learning of their VOR.

## Discussion

In the current study we demonstrate that triggering EPAC induces PKCε activation and threonine phosphorylation of RIM1α, which in turn facilitates the assembly of the Rab3A-RIM1α-Munc13-1 tripartite complex and thereby docking and release of synaptic vesicles at active zones of PF-PC synapses (*Figure 7—figure supplement 2*). The form of presynaptic LTP at these synapses that requires activation of the EPAC-PKCε module can be induced by either tetanic stimulation or forskolin at PF terminals (*Figure 7—figure supplement 2*). Via its presynaptic actions, the EPAC-PKCε module contributes to adaptation of compensatory eye movements, a motor learning task that depends on the vestibulo-cerebellum.

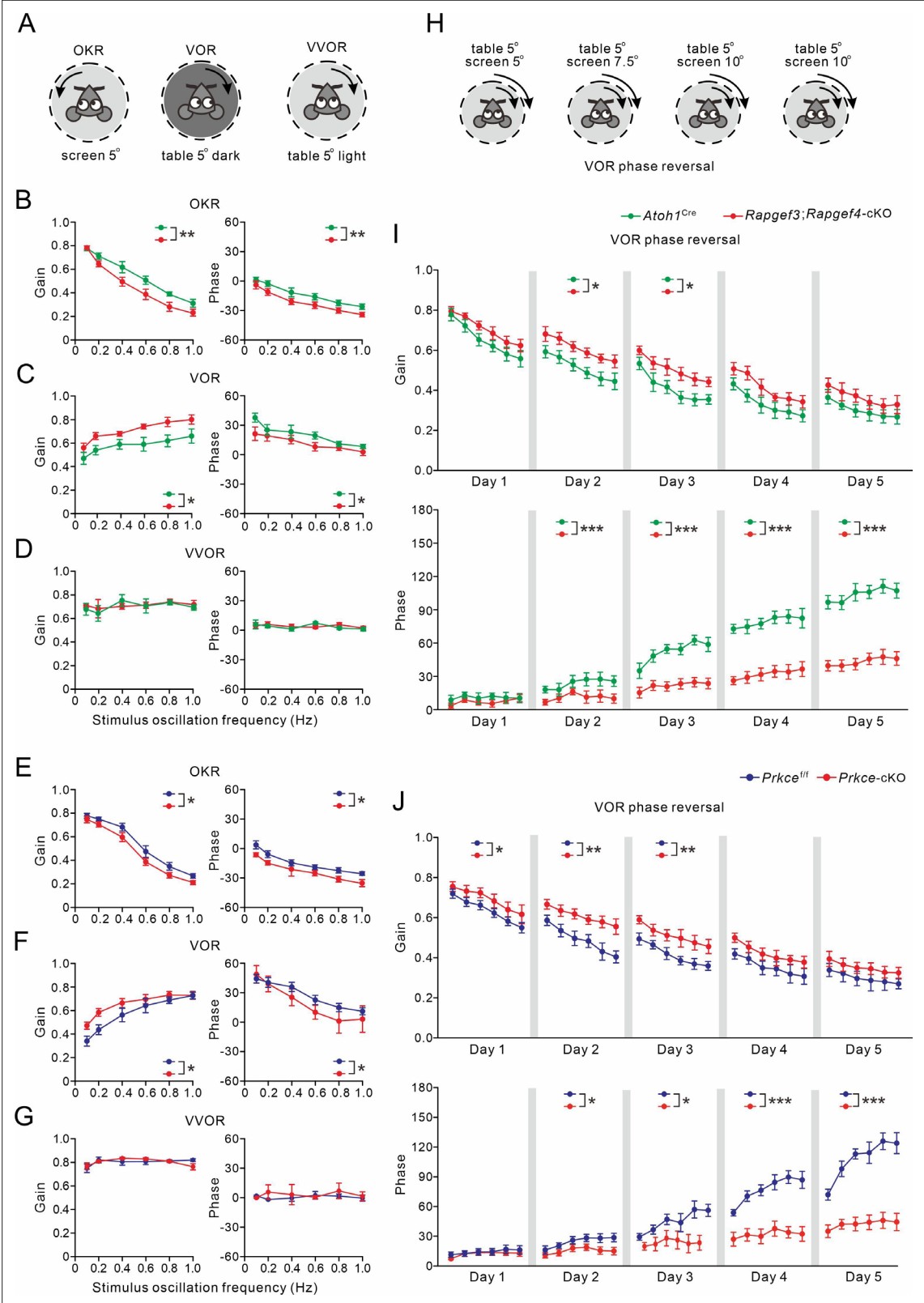

**Figure 7.** VOR baseline and adaptation in *Atoh1*^Cre, *Rapgef3;Rapgef4*-cKO, *Prkce*^f/f and *Prkce*-cKO mice. (**A**) Pictograms depicted compensatory eye movements driven by visual stimulus (OKR), vestibular stimulus (VOR) or both (VVOR). (**B**) OKR gain (measure of eye movement amplitude) and phase (measure of timing) were smaller in *Rapgef3;Rapgef4*-cKO (n=16) mice compared to *Atoh1*^Cre (n=10) mice. (**C**) VOR was affected in *Rapgef3;Rapgef4*-cKO mice. (**D**) The combination of vestibular and visual input by rotation of the mouse in the light evoked the VVOR in *Atoh1*^Cre and *Rapgef3;Rapgef4*-

*Figure 7 continued on next page*

*Figure 7 continued*

cKO mice. (**E**) OKR gain and phase were smaller in *Prkce*-cKO (n=11) mice compared to *Prkce*[f/f] (n=10) mice. (**F**) VOR was affected in *Prkce*-cKO mice. (**G**) VVOR gain and phase in *Prkce*[f/f] and *Prkce*-cKO mice. (**H**) Mismatched visual and vestibular input was used to trigger adaptation of the eye movements in order to test motor learning ability. This training induced a reversal of VOR phase probed by VOR recordings in the dark. (**I**) Both gain-decrease learning and phase learning of *Rapgef3;Rapgef4*-cKO were impaired. *p<0.05. ***p<0.001. (**J**) Both gain-decrease learning and phase learning of *Prkce*-cKO were impaired. *p<0.05. **p<0.01. ***p<0.001.

The online version of this article includes the following figure supplement(s) for figure 7:

**Figure supplement 1.** Impaired VOR learning in *Rapgef3/4*-dKO mice.

**Figure supplement 2.** Proposed schematic model for the function of EPAC-PKCε module in presynaptic LTP and motor learning.

## Threonine phosphorylation of RIM1α by the EPAC-PKCε module

Our finding that the EPAC-PKCε module can phosphorylate RIM1α raises a simple but fascinating mechanistic concept that the phosphorylation level of RIM1α determines presynaptic release. RIM1α specifically interacts with a number of presynaptic proteins, such as Munc13-1, liprin-α and ELKS, so as to form a scaffold complex regulating homeostatic release of synaptic vesicles (*Sudhof, 2004*). RIM1α can be phosphorylated at two serine residues by PKA and CaMKII (*Lonart et al., 2003*; *Sun et al., 2003*), which promotes its interaction with 14-3-3 protein (*Sun et al., 2003*). The current work advances on this concept by showing that RIM1α can also be phosphorylated at its threonine sites by PKCε. Moreover, our data demonstrate the functional implication consequence of threonine phosphorylation of RIM1α at PF-PC synapses: it promotes the assembly of the Rab3A-RIM1α-Munc13-1 complex and is essential for the induction of presynaptic PF-PC LTP, suggesting that a fast switch between phosphorylation and dephosphorylation of RIM1α may regulate presynaptic potentials during dynamic synaptic events. This new mechanistic concept is in line with the notion that synaptic vesicle proteins, such as RIM1α, often exhibit stimulation-dependent changes in phosphorylation (*Kohansal-Nodehi et al., 2016*). It remains to be elucidated how threonine and serine phosphorylations of RIM1α may exert distinct downstream effects. For instance, one could speculate that the threonine loci of RIM1α lead to more prominent conformational changes, allowing RIM1α to bind to active zone proteins. Here, we simultaneously deleted EPAC1 and EPAC2 in the granule cells, leaving the question open whether one might dominate the modulation of Rim1α phosphorylation. Similarly, it would be of interest to investigate the role of Rap1, the presynaptically expressed substrate of EPAC a (*Yang et al., 2012*), on RIM1α phosphorylation.

## Distinct roles of EPAC and PKA at synapses

cAMP-mediated signaling pathways that are mediated by EPAC and PKA regulate a multitude of physiological and pathological processes (*Cheng et al., 2008*). EPAC shares homologous cAMP-binding domains with PKA, but also possesses domains absent in PKA, such as the Ras exchange motif, the Ras association domain, and the CDC25-homology domain (*Cheng et al., 2008*). Indeed, the specific domains endow EPAC and PKA with different and even opposite functions. For example, in contrast to PKA, EPAC can activate small GTPase Rap1 (*de Rooij et al., 1998*) and increase PKB phosphorylation (*Mei et al., 2002*). Our current work bolsters the differences, showing that EPAC can phosphorylate PKCε and RIM1α threonine sites at synapses. This highlights the question as to how EPAC and PKA operate in an integrated manner to control the net physiological effect of cAMP-signaling pathways at synapses. Some studies indicate that presynaptic potentiation depends predominantly on PKA (*Salin et al., 1996*; *Linden and Ahn, 1999*; *Lev-Ram et al., 2002*), whereas others advocate a more critical role for EPAC (*Kaneko and Takahashi, 2004*; *Fernandes et al., 2015*; *Martín et al., 2020*). Our results highlight that ablation of either EPAC or PKCε by itself is not sufficient to block forskolin-induced synaptic potentiation, but that supplementing this with a blockage of PKA causes a complete blockage. These results demonstrate that EPAC and PKA can conjunctively regulate synaptic potentiation. Even so, our results clarify that the impact of EPAC on cAMP-induced EPSC potentiation is dominant, as it has the strongest contribution to the forskolin-induced increase of EPSC amplitude. Alternatively, PKA warrants a minimum level of potentiation that may be required under particular circumstances when EPAC is not active.

## The EPAC-PKCε module regulates synaptic release and is required for presynaptic LTP

Our EM analysis shows that the number of docked vesicles at the PF terminals of *Rapgef3/4*-dKO and *Prkce*-cKO mutants is reduced, whereas the general structure of PF-PC synapses is unchanged. As the ablation of either EPAC or PKCε attenuated protein interactions in the Rab3A-RIM1α-Munc13-1 complex, which is required for the docking and priming of presynaptic vesicles (*Schoch et al., 2002*; *Sudhof, 2004*; *Ferrero et al., 2013*), the reduction in docked vesicles in *Rapgef3/4*-dKO and *Prkce*-cKO mice can be readily explained. In parallel with our observations at the ultrastructural level, we found that mice with presynaptic deletion of EPAC and PKCε displayed obvious defects in synaptic release at the electrophysiological level. Although early studies have shown that EPAC1 and EPAC2 are involved in synaptic release in the hippocampus and the cerebellum (*Yang et al., 2012*; *Zhao et al., 2013*), which was further strengthened by *Martín et al., 2020*, our finding that PKCε acts as the downstream effector of EPAC and regulates presynaptic release is novel. Furthermore, we demonstrate for the first time that presynaptic PKCε is required for presynaptic LTP at PF-PC synapses. These findings expand the repertoire of forms of PC plasticity that are driven by cAMP signaling.

The role of the cAMP-PKA cascade in presynaptic LTP has been extensively debated. Early studies claimed that PKA and RIM1α serine phosphorylation are critical for the induction of presynaptic LTP at PF-PC synapses (*Salin et al., 1996*; *Lonart et al., 2003*). However, this conclusion was challenged by follow-up studies, demonstrating that RIM1α-S413A mutant mice exhibit normal presynaptic LTP in both cerebellum and hippocampus (*Kaeser et al., 2008*; *Yang and Calakos, 2010*). In our opinion, a couple of caveats must be considered regarding the function of PKA in presynaptic LTP. First, cAMP analogs (Rp-8-CPT-cAMP-S and Sp-8CPT-cAMP-S) used in two studies advocating that PKA mediates presynaptic PF-PC LTP (*Salin et al., 1996*; *Lonart et al., 2003*) are able to regulate Rap1 (*Roscioni et al., 2009*), which is a direct substrate of EPAC (*de Rooij et al., 1998*). Therefore, these cAMP analogs may also act through the EPAC-PKCε module. Second, KT5720 at 10 μm, a concentration used by *Lonart et al., 2003*, can alter a range of protein kinases, including phosphorylase kinase, mitogen-activated protein kinase kinase, PKBα, glycogen synthase kinase 3β, as well as AMP-activated protein kinase (*Brushia and Walsh, 1999*; *Davies et al., 2000*; *Murray, 2008*). Thus, KT5720 at this concentration has numerous side-effects next to its ability to inhibit PKA. In contrast, our results derived from cell-specific mouse lines consistently converge on the concept that presynaptic PF-PC LTP depends on the EPAC-PKCε module. More specifically, our data demonstrate that repetitive 8 Hz PF stimulation increases the level of cAMP and consequently activates EPAC and PKCε, which in turn induces threonine phosphorylation of RIM1α, suggesting a phospho-switch machinery that can tune presynaptic PF-PC LTP.

Our finding that the EPAC-PKCε module is a central component for synaptic release and presynaptic LTP may not stand on its own. In fact, EPAC is involved in cellular processes like cell adhesion, cell-cell junction formation, exocytosis and secretion, cell differentiation, as well as cell proliferation (*Cheng et al., 2008*), while PKCε is necessary for sperm exocytosis in the testis (*Lucchesi et al., 2016*). Together, these lines of evidence suggest that the EPAC-PKCε module might be a widespread mechanism controlling not only synaptic release in nerve cells, but also granule secretion in endocrine or proliferating cells. In addition, *Gutierrez-Castellanos et al., 2017* showed that EPAC may regulate GluA3 conductance in PCs, suggesting that postsynaptic EPAC or PKCε may regulate the conductance of AMPA receptor subunits, and thereby postsynaptic LTP or LTD at PF-PC synapses.

## Role of presynaptic LTP in motor behavior

Many studies have explored the potential functional role of postsynaptic plasticity at PC synapses, in particular that of PF-PC LTP and PF-PC LTD (*Gao et al., 2012*; *Raymond and Medina, 2018*). The picture emerging from these studies is that postsynaptic PF-PC LTP and PF-PC LTD play an important role in forms of learning that are mediated by the so-called upbound and downbound modules (*De Zeeuw, 2021*). Whereas VOR adaptation is mainly mediated by upbound microzones in the vestibulo-cerebellum that increase the simple spike frequency during learning (*Gutierrez-Castellanos et al., 2017*; *Voges et al., 2017*), eyeblink conditioning is predominantly regulated by downbound microzones in lobule simplex that decrease simple spikes during learning (*ten Brinke et al., 2015*; *Wu et al., 2019*). Yet, what is the role of presynaptic LTP at PF-PC synapses? Even though it has been suggested more than a decade ago that the functional role of presynaptic plasticity at PF-PC synapses

during learning can be expected to align with that of postsynaptic plasticity (*Le Guen and De Zeeuw, 2010*), evidence has been largely lacking.

Here, we found that *Rapgef3;Rapgef4*-cKO and *Prkce*-cKO mice, which showed reduced PF-PC transmission and lack presynaptic LTP, exhibit deficits in basic motor performance, in the form of an affected OKR and VOR, as well as in gain-decrease and phase reversal learning of their VOR (*Figure 7—figure supplement 2*). Similarly, presynaptic ablation of EPACs or PKCε results in altered gain and phase values of their OKR and VOR. Interestingly, the impairments in OKR and VOR caused by deletion of EPAC1/EPAC2 or PKCε in granule cells were similar to those caused by global deletion of EPAC. This finding raises the possibility that presynaptic EPAC is in fact more critical for basic motor performance than postsynaptic EPAC. This possibility is compatible with previous work showing that mice with a PC-specific deletion of GluA3, which leads to a lack of postsynaptic LTP mediated by EPAC, have hardly any significant deficit in basic motor performance (*Gutierrez-Castellanos et al., 2017*). By the same argument, the contribution of presynaptic LTP to phase reversal learning might be more in line with that of postsynaptic PF-PC LTP in that *Rapgef3;Rapgef4*-cKO and *Prkce*-cKO mice showed similar deficits as PC-specific GluA3 knockouts. The prediction that the impact of presynaptic plasticity at PF-PC synapses during learning operates in a synergistic fashion with that of postsynaptic plasticity (*Le Guen and De Zeeuw, 2010*), does in this respect hold. Two caveats should be considered in the present studies. First, *Atoh1*[Cre]-induced deletion of EPAC or PKCε might affect the function of unipolar brush cells (UBCs), which are involved in cerebellar ataxias (*Kreko-Pierce et al., 2020*). However, we believe that the EPAC-PKCε module regulates VOR learning through presynaptic plasticity mechanism at PF-PC synapses rather than UBCs, in line with the observations in other granule-cell-specific mutations (*Galliano et al., 2013*; *Schonewille et al., 2021*). Second, presynaptic PF-PC LTP was performed in the cerebellar vermis in the present work, whereas VOR learning generally requires PC activity in the flocculus. Unfortunately, we found that PC-EPSCs in the flocculus were not suitable to record PC plasticity because they were unstable.

Although we observed only a difference in phase at the end of VOR phase reversal training, it should be noted that the gain was different on multiple days in both *Rapgef3;Rapgef4*-cKO and *Prkce*-cKO mice compared to their controls. VOR phase reversal training subjects to multiple days of changing training stimuli to test different aspects of adaptation. The first aim is to decrease the gain, followed by an increase in phase. Once the phase has increased above 120°, the gain will increase again (*Wulff et al., 2009*). Therefore, the initial decrease of gain followed by the late-stage increase presumably underlies the absence of differences in gain between control and mutant groups in days 4 and 5. This does not imply that presynaptic LTP is more essential for the phase than the gain, as VOR gain decrease is affected during the first day of training.

# Materials and methods

## Animals

Original breeding pairs of *Rapgef3/4*-dKO and *Atoh1*[Cre] mice were obtained from Youmin Lu (Huazhong University of Science and Technology, Wuhan, China) and Wei Mo (Xiamen University, Xiamen, China), respectively. *Rapgef3*[f/f], *Rapgef4*[f/f] and *Prkce*[f/f] mice were made by us with the assistance of GemPharmatech (Soochow, Jiangsu, China) and Nanjing Biomedical Research Institute (Nanjing, Jiangsu, China). The resulting offspring were genotyped using PCR of genomic DNA. Mice were kept at the Experimental Animal Center of Zhejiang University under temperature-controlled condition on a 12:12 hr light/dark cycle. All experiments were performed blind to genotypes in age-matched littermates of either sex.

## Antibodies and reagents

Antibodies against RIM1 (Cat# 140013, RRID:AB_2238250 and Cat# 140023, RRID:AB_2177807), Rab3 (Cat# 107011, RRID:AB_887768) and Munc13-1 (Cat# 126102, RRID:AB_887734) were from Synaptic Systems (Gottingen, Germany). Antibodies against phosphor-threonine (Cat# 9381, RRID:AB_330301), EPAC1 (Cat# 4155, RRID:AB_1903962) and EPAC2 (Cat# 4156, RRID:AB_1904112) were from Cell Signaling (Danvers, MA). The antibody to phosphor-serine (Cat# AB1603, RRID:AB_390205) was from Millipore (Billerica, MA). Antibodies against HA (Cat# M20003, RRID:AB_2864345), Flag (Cat# M20008, RRID:AB_2713960) and His (Cat# M30111, RRID:AB_2889874) were from Abmart (Shanghai, China).

Antibody against PKCα (Cat# P4334, RRID:AB_477345) was from sigma (St. Louis, MO). Antibodies against PKCα-pS657 (ab23513, RRID:AB_2237450), PKCα-pT638 (Cat# ab32502, RRID:AB_777295), PKCε-pSer729 (Cat# ab63387, RRID:AB_1142277), EPAC1 (Cat# ab21236, RRID:AB_2177464, for immunostaining), EPAC2 (Cat# ab124189, RRID:AB_10974926, for immunostaining), anti-mouse IgG for IP (HRP) (Cat# ab131368, RRID:AB_2895114) and VeriBlot for IP Detection Reagent (HRP) (Cat# ab131366, RRID:AB_2892718) were from Abcam (Cambridge, UK). Antibody against β-tubulin (Cat# sc-5274, RRID:AB_2288090) was from Santa Cruz (Dallas, TX). Antibody against PKCε (Cat# MA5-14908, RRID:AB_10985232), Goat anti-mouse IgG horseradish peroxidase (HRP)-conjugated (Cat# 31446, RRID:AB_228318), Goat anti-rabbit IgG horseradish peroxidase (HRP)-conjugated (Cat# 31460, RRID:AB_228341) were from Thermo Fisher Scientific (Waltham, MA). Anti-vGluT1 antibody was a gift from Dr. Masahiko Watanabe (Hokkaido University, Sapporo, Japan). The antibody against PKCε (Cat# 20877–1-AP, RRID:AB_10697812, for immunostaining) was from Proteintech (Rosemont, IL). Mouse IgG (Cat# A7028, RRID:AB_2909433) and rabbit IgG (Cat# A7016, RRID:AB_2905533) were from Beyotime (Shanghai, China). Protease inhibitor cocktail (04693132001) was from Roche (Mannheim, Germany). Gö6976 (2253), 8-pCPT (4853) and FR236924 (3091) were from Tocris (Bristol, UK). Dulbecco's modified Eagle's medium (DMEM, 11885–084), Penicillin-Streptomycin (15140–122), Sodium Pyruvate (11360–070), Fetal Bovine Serum (FBS, 10099–133), lipofectamine 2000 (11668–019), OPTI-MEM (31985–062), and Alexa Fluor-conjugated secondary antibodies were from Invitrogen (Carlsbad, CA). GammaBind Plus Sepharose (17-0886-01) was from GE healthcare. Other chemicals were from Sigma (St. Louis, MO) unless stated otherwise.

## Plasmid construction

The construction of plasmids was performed according to previous work (*Zhou et al., 2015*). HA-RIM1α, Flag-EPAC1, Flag-EPAC2, and His-PKCε, were constructed based on the coding sequence of rat *Rim1a* gene (GenBank# NM_052829.1), rat *Rapgef3* gene (GenBank# NM_021690.1), rat *Rapgef4* gene (GenBank# XM_017592164.1), and rat *Prkce* gene (GenBank# NM_017171.1), respectively. All constructs were verified by DNA sequencing.

## RT-PCR

RT-PCR was used to determine the mRNA level of EPAC1, EPAC2 and PKCε in granule cells. The contents of individual granule cells (P21) were harvested as described in previous work (*Zhou et al., 2017*). The harvested contents were subjected to RT-PCR using OneStep Kit (210212, Qiagen, Hilden, Germany). Forward (F) and reverse (R) primers used for amplification were as follows: *Rapgef3*, F: 5'-GCT TGT TGA GGC TAT GGC-3'; R: 5'- ACA CAG TTC CTG CCT TGC-3'. *Rapgef4*, F: 5'- CAT TCT CTC TCG AGC TCC-3'; R: 5' TGG TTG AGG ACA CCA TCT-3'. *Prkce*, F: 5'- ATT GAC CTG GAG CCA GAA –3'; R: 5'- CTT GTG GCC ATT GAC CTG-3'. *Gapdh*, F: 5'-GGT GAA GGT CGG TGT GAA CG-3 '; R: 5'-CTC GCT CCT GGA AGA TGG TG-3'.

## HEK cell culture

HEK cells were cultured in DMEM supplemented with 10% FBS, 1 mM sodium pyruvate, 100 U/ml penicillin, and 10 μg/ml streptomycin and stored in an incubator (95% $O_2$/5% $CO_2$; 37 °C). The plasmids were transfected to HEK cells in OPTI-MEM using lipofectamine 2000 (Invitrogen) at 70–80% confluency.

## Purification of synaptosomes

Synaptosomes were purified according to previous work (*Ferrero et al., 2013*). Cerebellar tissues from mice (P21) were homogenized in a medium (pH7.4) containing sucrose (320 mM) and protease inhibitors. The homogenate was centrifuged 2000×*g* (4 °C for 2 min) and the supernatant was spun again at 9500×*g* (4 °C for 12 min). The compacted white layer containing the majority of synaptosomes was gently resuspended in sucrose (320 mM) supplemented with protease inhibitors, and an aliquot of synaptosomal suspension (2 ml) was placed onto a 3 ml Percoll discontinuous gradient (GE Healthcare) containing (in mM) 320 sucrose, 1 EDTA, 0.25 DL-dithiothreitol, and 3, 10, or 23% Percoll. After centrifugation at 25,000×*g* (4 °C for 10 min), synaptosomes were recovered from between 10% and 23% bands and diluted in a medium (in mM) (140 NaCl, 5 KCl, 5 NaHCO₃, 1.2 NaH₂PO₄, 1 MgCl₂,

10 glucose, 10 HEPES; pH 7.4) supplemented with protease inhibitors. The synaptosomes good for experiments were harvested from the pellet after the final centrifugation at 22,000×$g$ (4 °C for 10 min).

## Immunocytochemistry

For immunocytochemistry, synaptosomes were added to a medium containing 0.32 M sucrose (pH 7.4), allowed to attach to polylysine-coated coverslips for 1 hr, and fixed for 10 min in 4% paraformaldehyde in 0.1 M phosphate buffer (PB) (pH 7.4) at room temperature. Following several washes with PB (pH 7.4), synaptosomes were incubated for 1 hr in 10% normal goat serum diluted in PBS (pH 7.4) containing 0.2% Triton X-100. Subsequently, they were incubated for 24 hr with primary antiserum for EPAC1 (1:500), EPAC2 (1:500), PKCε (1:500) and vGluT1 (1:500). After washing in PBS, synaptosomes were incubated with secondary antibodies for 2 hr. Coverslips were mounted with Prolong Antifade Kit (Molecular Probes) and synaptosomes were viewed using a confocal microscope (Nikon A1R) with a×100 objective.

## Co-immunoprecipitation

After measuring protein concentration using the BCA assay, a tenth of lysis supernatant derived from synaptosomes or cultured cells was used for input and the remainder were incubated with anti-RIM1 or anti-HA antibody, which was precoupled to GammaBind Plus Sepharose at 5–10 μg antibody/1 ml beads for 3 hr. Proteins on the beads were extracted with 2×SDS sample buffer plus protease inhibitors and boiled for 5 min before western blot.

## Western blotting

The protein concentration was determined using BCA protein assay. Equal quantities of proteins were loaded and fractionated on SDS-PAGE, transferred to PVDF membrane (Immobilon-P, Millipore), immunoblotted with antibodies, and visualized by enhanced chemiluminescence (Thermo Fisher). The dilutions of primary antibodies were 1:1,000 for RIM1, Munc13-1, PKCα-pS657, EPAC1, EPAC2, p-Thr, p-Ser, β-tubulin, and PKCε-pSer729; 1:2,000 for Rab3A and PKCε; 1:5,000 for PKCα-pT638; 1:10,000 for HA, His, Flag, GAPDH, and PKCα. Secondary antibodies were goat anti-rabbit (1:10,000), goat anti-mouse (1:10,000), anti-mouse IgG for IP (HRP) (1:1,000), VeriBlot for IP Detection Reagent (HRP) (1:1,000). Film signals were digitally scanned and quantified using ImageJ 1.42q (NIH, Bethesda, MD).

## Electron microscopy

After anesthetic mice (P21) were transcardially perfused with saline and ice-cold fixative, brains were removed and stored at 4 °C for 2.5 hr in fixative. Sagittal slices of vermis (200 μm) were prepared and rectangular molecular layer sections from lobules IV-V were dissected. The samples were dehydrated and embedded in an epoxy resin. Ultrathin sections (90 nm) were cut using an ultra-microtome (Leica), stained with 2% uranyl acetate and lead solution, and mounted on grids. EM images were captured at ×30,000 magnification using a Tecnai transmission electron microscope (FEI, Hillsboro, OR). PF-PC synapses were identified by asymmetrical and short contacts, which were distinct from GABAergic or climbing fiber synapses (*Ichikawa et al., 2016*). ImageJ was used to count the numbers of total and docked vesicles per bouton.

## Golgi staining and spine density analysis

Golgi staining was performed using Rapid Golgi Stain Kit (FD NeuroTech Inc, Ellicott, MD) according to the manufactory's instruction. PCs at the apical region were imaged using a bright field microscope (Zeiss, Germany). ImageJ was used to count the spine number and dendrites length of PCs with manual assistant.

## Electrophysiology

Sagittal slices of cerebellar vermis (250 μm) were prepared from anesthetic mice (P21) using a vibrating tissue slicer (Leica VT1000S) and ice-cold standard artificial cerebrospinal fluid (aCSF) containing (in mM): 125 NaCl, 2.5 KCl, 1.25 NaH$_2$PO$_4$, 1 MgCl$_2$, 2 CaCl$_2$, 26 NaHCO$_3$ and 25 D-glucose, bubbled with 95% O$_2$/5% CO$_2$. When low Ca$^{2+}$ (0.5 mM) was used, Mg$^{2+}$ concentration was increased to 2.5 mM. After recovery for 30 min at 37 °C, slices were placed in a submerged chamber that was perfused at 2 ml/min with aCSF supplemented with GABAzine (10 μM) during recordings.

PCs were visualized under an upright microscope (BX51, Olympus) equipped with a 40×water-immersion objective and infrared differential interference contrast enhancement. Whole-cell recordings were made on PCs from lobules IV-V with a MultiClamp 700B amplifier (Molecular Devices). Currents were digitized at 10 kHz and filtered at 3 kHz. Patch electrodes (3–5 MΩ) were filled with an intracellular solution containing (in mM) 135 Cs-methanesulfonate, 10 CsCl, 10 HEPES, 0.2 EGTA, 4 $Na_2ATP$, and 0.4 $Na_3GTP$ (pH 7.3, OSM 290). PCs were held at –70 mV to prevent spontaneous spikes that might escape clamp. For PF stimulation, standard patch pipettes were filled with aCSF and placed in middle third of molecular layer. Presynaptic PF-PC LTP was induced by stimulating PF input 120 times at 8 Hz (*Salin et al., 1996*; *Kaeser et al., 2008*). Postsynaptic PF-PC LTP was obtained when PFs were stimulated at 1 Hz for 5 min in parallel with current-clamp of recording PC (*Wang et al., 2014*). PF-LTD was induced by a conjunction of 5 PF-pulses at 100 Hz and a 100 ms long depolarization of PC to 0 mV, which was repeated 30 times with an interval of 2 s (*Zhou et al., 2015*). mEPSCs were recorded in whole-cell configuration in the presence of tetrodotoxin (0.5 μM) and an offline analysis was conducted using a sliding template algorithm (ClampFit 10, Molecular Device) according to previous work (*Zhou et al., 2017*). To estimate RRP and Pr, a repeated 100 Hz train stimulation protocol was used to evoke 50 EPSCs. RRP was calculated by linear interpolating the linear portion of the cumulative EPSC amplitude plot to virtual stimulus 0. Pr was calculated as the normalized 1st EPSC during the train stimulations divided by RRP (*Thanawala and Regehr, 2016*; *He et al., 2019*). A temperature controller was used to elevate aCSF temperature in the recording chamber (TC-344C; Warner Instruments, Holliston, MA).

## Compensatory eye movement test

Mice (P60) were surgically prepared for head-restrained recordings of compensatory eye movements. A pedestal was attached to the skull after shaving and opening the skin overlaying it, using Optibond primer and adhesive (Kerr, Bioggio, Switzerland) and under isoflurane anesthesia in O2 (induction with 4% and maintained at 1.5% concentration). Mice were administered xylocaine and an injection with bupivacaine hydrochloride (2.5 mg/ml, bupivacaine actavis) to locally block sensation. The pedestal consisted of a brass holder (7×4 mm base plate) with a neodymium magnet (4×4 × 2 mm) and a screw hole for fixation. After a recovery period of at least 3 days, mice were placed in a mouse holder, using the magnet and a screw to fix the pedestal to a custom-made restrainer, and the mouse was placed with the head in the center on a turntable (diameter 60 cm) in the experimental setup. A drum (diameter 63 cm) surrounded the mouse during the experiment. The recording camera was calibrated by moving the camera left–right by 20° peak to peak at different light levels. Compensatory eye movement performance was examined by recording the OKR, VOR, and VVOR using a sinusoidal rotation of the drum in light (OKR), rotation of the table in the dark (VOR), or rotation of the table (VVOR) in the light. These reflexes were evoked by rotating the table and/or drum at 0.1–1 Hz (20–8 cycles, each recorded twice) with a fixed 5° amplitude. In order to evaluate motor learning, a mismatch between visual and vestibular input was used to adapt the VOR. The ability to perform VOR phase reversal was tested using a 5 day paradigm, consisting of six 5 minute training sessions every day with VOR recordings before, between, and after the training sessions. On the first day during training, the visual and vestibular stimuli rotated in phase at 0.6 Hz and at the same amplitude, inducing a decrease of gain. On the subsequent days, the drum amplitude was increased relative to the table and induced the phase reversal of the VOR, resulting in a compensatory eye movement in the same direction as the head rotation instead of the normal compensatory opposite direction (all days vestibular 5° rotation, visual day 2: 5°; day 3, 7.5°; days 4–5, 10°). Between recording sessions, mice were kept in the dark to avoid unlearning of the adapted responses.

Eye movements were recorded with a video-based eye-tracking system (hard- and software, ETL-200; ISCAN systems, Burlington, MA). Recordings were always taken from the left eye. The eye was illuminated during the experiments using two table-fixed infrared emitters (output 600 mW, dispersion angle 7°, peak wavelength 880 nm) and a third emitter that was mounted to the camera, aligned horizontally with the optical axis of the camera, which produced the tracked corneal reflection. Pupil size and corrected (with corneal reflection) vertical and horizontal pupil positions were determined by the ISCAN system, filtered (CyberAmp; Molecular Devices, San Jose, CA), digitized (CED, Cambridge, UK) and stored for offline analysis. All eye movement signals were calibrated, differentiated to obtain velocity signals, and high-pass–filtered to eliminate fast phases, and then cycles were averaged.

Gain—the ratio of eye movement amplitude to stimulus amplitude—and phase values—time difference between eye and stimulus expressed in degrees—of eye movements were calculated using custom-made MATLAB routines (The MathWorks, Natick, MA).

## Statistical analysis

Experimenters who performed experiments and analyses were blinded to the genotypes until all data were integrated. Data were analyzed using Igor Pro 6.0 (Wavemetrics, Lake Oswego, OR), Graphpad Prism (San Diego, CA), SPSS 16.0 (IBM, Chicago, IL), and MATLAB. No statistical methods were used to pre-determine sample sizes, which were based on our previous studies. All data sets were tested for the assumptions of normality of distribution. No data were excluded except electrophysiological recordings with ≥15% variance in series resistance, input resistance, or holding current. Standard deviations for control were calculated from the average of all control data. Statistical differences were determined using unpaired or paired two-sided Student's $t$ test for two-group comparison, or one-way ANOVA followed by Tukey's post hoc test for multiple comparisons, or repeated measures ANOVA for repeated measures. The accepted level of significance was $p < 0.05$. '$n$' represents the number of preparations or cells. Data in the text and figures are presented as mean ± SEM.

## Acknowledgements

We thank Drs. Jia-Dong Chen and Yan Gu for their comments on this work, Drs. You-Min Lu and Wei Mo for providing *Rapgef3/4*-dKO and *Atoh1*[Cre] mice, Dr. Lan Bao for providing assistance in electron microscopy, Drs. Min Wu and Jun Xia for their participation at the beginning of this work, and the Core Facilities of Zhejiang University Institute of Neuroscience for technical assistance. This work was supported by grants from National Innovation of Science and Technology-2030, STI2030-Major Projects 2021ZD0204000 (to YS), National Natural Science Foundation of China (81625006 to YS, 31820103005 to YS, 32000692 to XTW, 32160192 to YW, 32100791 to FXX, 31900741 to LZ, 32170976 to LZ), National Key Research and Development Program of the Ministry of Science and Technology of China (2020YFB1313500 to LZ), Science and Technology Innovation Commission of Shenzhen Municipal Government (JCYJ20160331115633182 to SJJ), Science and Technology Programme of Hangzhou Municipality (20190101 A10 to WC), Key Realm R&D Program of Guangdong Province (2019B030335001 to WC), Ningxia Key Research and Development Program (2021BEG03097 to YW), Natural Science Foundation of Zhejiang Provincea (LQ17C090001 to NW), ERC-Stg (680235 to MS), Dutch Organization for Medical Sciences (CIDZ) and Life Sciences (CIDZ), and ERC-adv and ERC-POC of the EU (CIDZ), INTENSE (CIDZ), and NIN Vriendenfonds for albinism (CIDZ).

## Additional information

### Funding

| Funder | Grant reference number | Author |
| --- | --- | --- |
| National Innovation of Science and Technology-2030 | 2021ZD0204000 | Ying Shen |
| National Natural Science Foundation of China | 81625006 | Ying Shen |
| National Natural Science Foundation of China | 31820103005 | Ying Shen |
| National Natural Science Foundation of China | 32000692 | Xin-Tai Wang |
| National Natural Science Foundation of China | 32160192 | Yin Wang |
| National Natural Science Foundation of China | 32100791 | Fang-Xiao Xu |

| Funder | Grant reference number | Author |
|---|---|---|
| National Natural Science Foundation of China | 31900741 | Lin Zhou |
| National Natural Science Foundation of China | 32170976 | Lin Zhou |
| National Key Research and Development Program of China | 2020YFB1313500 | Lin Zhou |
| Science, Technology and Innovation Commission of Shenzhen Municipality | JCYJ20160331115633182 | Sheng-Jian Ji |
| Science and Technology Programme of Hangzhou Municipality | 20190101A10 | Wei Chen |
| Key Realm R&D Program of Guangdong Province | 2019B030335001 | Wei Chen |
| Ningxia Key Research and Development Program | 2021BEG03097 | Yin Wang |
| Natural Science Foundation of Zhejiang Province | LQ17C090001 | Na Wang |
| ERC-Stg | 680235 | Martijn Schonewille |
| Dutch Organization for Medical Sciences | | Chris I De Zeeuw |
| Dutch Organization for Life Sciences | | Chris I De Zeeuw |
| ERC-adv and ERC-POC of the EU | | Chris I De Zeeuw |
| INTENSE | | Chris I De Zeeuw |
| NIN Vriendenfonds for albinism | | Chris I De Zeeuw |

The funders had no role in study design, data collection and interpretation, or the decision to submit the work for publication.

## Author contributions

Xin-Tai Wang, Conceptualization, Data curation, Formal analysis, Funding acquisition, Validation, Investigation, Methodology, Writing – original draft, Writing – review and editing; Lin Zhou, Conceptualization, Data curation, Formal analysis, Funding acquisition, Investigation, Methodology, Writing – original draft; Bin-Bin Dong, Data curation, Formal analysis, Investigation; Fang-Xiao Xu, Conceptualization, Data curation, Formal analysis, Funding acquisition, Validation, Investigation, Methodology, Writing – original draft; De-Juan Wang, Data curation, Formal analysis, Validation, Investigation, Visualization, Methodology; En-Wei Shen, Xin-Yu Cai, Data curation, Formal analysis, Investigation, Methodology; Yin Wang, Funding acquisition, Validation, Investigation, Methodology, Writing – original draft; Na Wang, Data curation, Formal analysis, Methodology; Sheng-Jian Ji, Funding acquisition, Validation, Visualization, Writing – original draft; Wei Chen, Resources, Funding acquisition; Martijn Schonewille, Funding acquisition, Validation, Visualization, Writing – original draft, Writing – review and editing; J Julius Zhu, Conceptualization, Supervision, Writing – original draft, Project administration, Writing – review and editing; Chris I De Zeeuw, Supervision, Funding acquisition, Validation, Visualization, Project administration, Writing – review and editing; Ying Shen, Conceptualization, Supervision, Funding acquisition, Visualization, Methodology, Writing – original draft, Project administration, Writing – review and editing

## Author ORCIDs

Na Wang http://orcid.org/0000-0002-1438-1508
Sheng-Jian Ji http://orcid.org/0000-0003-3380-258X

Martijn Schonewille http://orcid.org/0000-0002-2675-1393
J Julius Zhu http://orcid.org/0000-0002-1879-983X
Chris I De Zeeuw http://orcid.org/0000-0001-5628-8187
Ying Shen http://orcid.org/0000-0001-7034-5328

### Ethics

All of the animals were handled according to approved protocol of the Animal Experimentation Ethics Committee of Zhejiang University (ZJU17067).

### Decision letter and Author response

Decision letter https://doi.org/10.7554/eLife.80875.sa1
Author response https://doi.org/10.7554/eLife.80875.sa2

## Additional files

### Supplementary files

• Transparent reporting form

### Data availability

All data generated or analysed during this study are included in the manuscript and supporting files; Source Data files have been provided for Figures 1, 2, and Figure 1-figure supplement 2, 3, and 4.

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
