## [Editor Report]

The cerebellum plays a critical role in motor learning, but exactly which forms of synaptic plasticity contribute to learning and the underlying molecular mechanisms remain poorly understood. In this study, Wang and colleagues show that presynaptic long-term potentiation at the parallel fiber to Purkinje cell synapse is required for one form of motor learning, and involves a previously-unknown signaling cascade, where EPAC activation leads to PKCε-dependent threonine phosphorylation of RIM1α. The evidence is compelling and convincing. This study provides fundamental and new insights into the underlying mechanisms and functional consequences of presynaptic LTP.

---

## [Decision Letter]

**Decision letter after peer review:**

Thank you for submitting your article "cAMP-EPAC-PKCε-RIM1α signaling regulates presynaptic long-term potentiation and motor learning" for consideration by *eLife*. Your article has been reviewed by 3 peer reviewers, and the evaluation has been overseen by a Reviewing Editor and Lu Chen as the Senior Editor. The reviewers have opted to remain anonymous.

Essential revisions:

1) 1. The stimulation protocol for estimating the RRP is not sufficient – The authors need to provide additional experimental data.

2) Please conduct additional control experiments – apply KT5720 on control animals to look at the effect.

3) The cumulative EPSC analysis in its present form is confusing. Please plot RRP size instead of commutative EPSC values.

4) Substantial revision of the test is recommended – Please follow the reviewers' detailed suggestions below, such as tuning down the conclusion, providing a better rationale in the introduction, and adding a discussion on the difference in experimental conditions between ex vivo and in vivo experiments, and lastly provide more citations of the prior publications.

*Reviewer #1 (Recommendations for the authors):*

1. My primary concern is the manner in which the authors choose to refer to previous studies, which seem designed to highlight the novelty of their work rather than provide context and information for the reader. For example, Line 44 "However, the molecular underpinnings of presynaptic plasticity in the cerebellar cortex are merely starting to be explored (Wang et al., 2021a)" is a baffling statement, given that the molecules involved in presynaptic LTP at this synapse have been described since the 1990s in studies like Salin et al., 1996 (cited by the authors), and Storm et al. 1998 (uncited).

Line 45: "whether presynaptic plasticity plays a role in cerebellar motor learning remains to be elucidated". Again, Storm et al. 1998 showed impaired rotarod performance and learning in AC1 knockout mice. The genetic manipulations in the current study are far cleaner than Storm et al., and the physiology and behavior are more convincing, but they should not minimize the impact of past studies.

The introduction and beginning of the results provide no context for the authors' decision to study EPAC. I can only assume that the results of Martin et al. 2020 provided a compelling rationale for this study, and think it should be included in the introduction.

2. Figure 1C: The statement that ENACs bind to RIM would be more convincing if the authors provide a negative control showing that the assay does not detect another presynaptic protein that does not interact with RIM does not appear in the pulldown. That said, interactions between ENAC and RIM have been reported before, and those studies could be cited.

3. Figure 2E, F: I have multiple concerns with this data. First, the RRP is calculated from EPSC trains collected evoked by only 40 stimuli. 40 stimuli are insufficient to reach a steady state at PF-PC synapses, which is evident in the example EPSC traces. Without reaching a steady state, it is impossible to estimate the resupply rate. Thus the RRP measurements are not accurate. Second: Without an accurate measure of the RRP, the initial release probability cannot be determined. Finally, although Line 166 states that the RRP was significantly reduced in the mutant mice, that data is not presented in the text of the figure. Instead, the figure shows the "cumulative EPSC", which is highly dependent on the number of presynaptic terminals that were stimulated. The average RRP sizes, measured by back extrapolation, actually appear to be very similar across genotypes. The authors should not make claims about release probability or RRP unless they repeat these experiments with longer stimulus trains that reach steady state amplitudes, and the RRP is normalized to the size of the initial EPSC in each experiment.

4. Line 240: "However, it remains unclear which downstream effector, EPAC or PKA (Cheng et al., 2008), is responsible for cAMP-induced potentiation." Martin et al. 2020 present compelling evidence that presynaptic LTP is driven by EPAC at the PF-PC synapse. This study is cited obliquely, and should not be ignored to enhance the novelty of the current study. For example, Line 433 understates the relevance of Martin to the current study: "EPAC1 and EPAC2 have been shown to be involved in synaptic release in the hippocampus and cerebellum".

5. The rationale for proceeding immediately to EPAC1/2 DKOs was not clear, leaving the reader to guess the reason? Are EPAC1/2 redundant in the EPAC-Rim1 pathway, or does one isoform dominate in potentiating this synapse? Could the authors make it clear why all experiments were done with DKOs?

6. Figure 4: What concentration of KT5720 was used to inhibit PKA? The authors note on Line 441 that previous studies used 10 microM, a concentration that produces off-target effects, but I cannot find the concentration the authors used. From their experiments, the authors conclude that the EPAC-PKC module is dominant in driving forskolin-induced potentiation. However, an obvious experiment would have been to apply KT to the control animals (Math1-Cre). This would rule out the possibility that PKA is activated somewhere within the EPAC-PKC, and plays a major role in potentiation.

7. The authors hypothesize that PLC is involved in this pathway (line 93 and Figure 1). They might use the discussion to highlight which aspects of their signaling pathway remain to be experimentally verified in presynaptic LTP.

*Reviewer #2 (Recommendations for the authors):*

The introduction part should elaborate more on the rationale behind the selection of EPAC to conduct the study of presynaptic plasticity.

The activities of EPAC and PKCε required for presynaptic LTP are shown. However, the change of the basal synaptic properties due to the knockdown of two proteins is not shown, such as the input-output relation of evoked EPSC in the PF-PC synapse. Therefore, studying the protein during the normal condition may be of interest.

The visual diagram (Figure 1G) of the authors' hypothesis on a process describing EPAC-mediated phosphorylation of RIM1 is a bit difficult to follow. Arrows may be indicated with more comprehensible signs/ words.

Following the compensatory eye movement training, the deficit is more prominent in phase than gain. Mutants catch up the gain at the end of 5 days of training but not the phase. The authors may explain how the presynaptic LTP is more essential to phase in the discussion. In addition, the reason to choose VOR phase reversal learning other than OKR learning or VOR gain-up learning may be related to this discussion.

In the discussion part, some more thoughts and scientific predictions of the limited role of EPAC and PKCε on postsynaptic plasticity should be stated.

Overall, specific claims made by the authors were straightforward and supported by an ample amount of experimental results. This study has widened the spectrum of understanding of the molecular complexity of cerebellar motor learning by demonstrating the active contribution of presynaptic plasticity and its related signaling cascade.

*Reviewer #3 (Recommendations for the authors):*

The conclusions often overstate the findings of the data. The language can be toned-down a without diminishing the importance of these findings.

In the present form, I don't think the cumulative EPSC analysis is informative. The authors should repeat the analysis using the Elmqvist and Quastel (EQ) method (see Thanawala and Regher, 2016). This method is more reliable at low Pr and can provide (when combined with the Train method) a high and low bracket of possible Pr and RRP values. Ideally, the authors would redo these experiments with a higher stimulus frequency. It would also be helpful to show a simple PPR value for each genotype. This can be done without additional experiments.

Slice conditions in LTP/LTD experiments (age, location, temperature, Ca^2+^ concentration) do not match in vivo conditions. These differences and potential changes in synaptic plasticity should be discussed.

Is it possible to provide an estimate of what percent of synaptosomes arise from parallel fibers versus mossy fibers (which also express VGLUT1) based on the relative number of each synapse? This would help the interpretation of the synaptosome data.

---

## [Author Response]

Essential revisions:1) 1. The stimulation protocol for estimating the RRP is not sufficient – The authors need to provide additional experimental data.

We have included new data using a new 100-Hz train protocol to estimate RRP and Pr, according to the reviewers’ suggestions. These results have been added in the updated text and figure legends (Figure 2E and 2F), and the corresponding methods have also been updated.

2) Please conduct additional control experiments – apply KT5720 on control animals to look at the effect.

We have conducted new experiments to investigate the effect of KT5720 on forskolin-induced potentiation. The results have been presented in Figure 4—figure supplement 1. We found that KT5720 (3 μM) inhibited the forskolin-induced potentiation by 15%, which is significant but much smaller than EPAC effect.

3) The cumulative EPSC analysis in its present form is confusing. Please plot RRP size instead of commutative EPSC values.

The related sentences have been rewritten in text and figure legend. We have measured RRP and Pr using an additional protocol (100-Hz train; see Thanawala and Regher, 2016) as the reviewers suggested. Under this condition, the RRPs of control and cKO mice were similar (*Math1*-Cre: 861 ± 113 vs *Epac1^cKO^;Epac2^cKO^* 790 ± 101, *p* = 0.31; Prkce^f/f^ 764 ± 100 vs Prkce^cKO^ 728 ± 106, *p* = 0.40). These results have been included in new Figure 2E and 2F.

4) Substantial revision of the test is recommended – Please follow the reviewers' detailed suggestions below, such as tuning down the conclusion, providing a better rationale in the introduction, and adding a discussion on the difference in experimental conditions between ex vivo and in vivo experiments, and lastly provide more citations of the prior publications.

We have tuned down the language, provided the rationale in the introduction, revised the discussion, and added more citations as suggested. Please see our detailed answers to each comment.

Reviewer #1 (Recommendations for the authors):1. My primary concern is the manner in which the authors choose to refer to previous studies, which seem designed to highlight the novelty of their work rather than provide context and information for the reader. For example, Line 44 "However, the molecular underpinnings of presynaptic plasticity in the cerebellar cortex are merely starting to be explored (Wang et al., 2021a)" is a baffling statement, given that the molecules involved in presynaptic LTP at this synapse have been described since the 1990s in studies like Salin et al., 1996 (cited by the authors), and Storm et al. 1998 (uncited).

We apologize for the confusions. In the original manuscript, we intended to make the introduction more concise. This choice resulted in a simple, short introduction that lacked relevant contextual information. We have re-written the introduction to present current status of presynaptic LTP at PF-PC synapses. In particular, the sentences and references that the reviewer indicated have been modified and added, respectively. “Relatively speaking, the molecular underpinnings of presynaptic plasticity in the cerebellar cortex are less understood (Wang et al., 2021a), though early studies have shown that presynaptic Ca influx, Ca-sensitive adenylate cyclase, and cyclic adenosine monophosphate (cAMP) production are required for presynaptic LTP (Byrne and Kandel, 1996; Salin et al., 1996; Storm et al., 1998)” (Line 45).

Line 45: "whether presynaptic plasticity plays a role in cerebellar motor learning remains to be elucidated". Again, Storm et al. 1998 showed impaired rotarod performance and learning in AC1 knockout mice. The genetic manipulations in the current study are far cleaner than Storm et al., and the physiology and behavior are more convincing, but they should not minimize the impact of past studies.

We have re-written the introduction to describe the prior knowledge regarding the relationship between presynaptic plasticity and cerebellar motor learning in more details. “Moreover, the function of presynaptic plasticity on cerebellar motor learning remains to be elucidated (Le Guen and De Zeeuw, 2010; De Zeeuw, 2021), though it was suggested that adenylyl cyclase-dependent LTP participates in rotarod learning (Storm et al., 1998)” (Line 49).

The introduction and beginning of the results provide no context for the authors' decision to study EPAC. I can only assume that the results of Martin et al. 2020 provided a compelling rationale for this study, and think it should be included in the introduction.

This is an excellent comment. We actually started this project in 2014, and finished part of current data and submitted the manuscript entitled “EPAC-PKCε Module Induces Threonine Phosphorylation of RIM1 and Presynaptic Long-Term Potentiation” to Neuron on August 2018. In that cover letter of Neuron submission, we wrote “To our knowledge, the present work provides compelling evidence to answer two major neuroscientific questions: (1) Compared to postsynaptic plasticity, the molecular underpinnings for the plastic changes at presynaptic side are poorly understood. It is shown that presynaptic LTP requires cAMP and scaffold protein RIM1α. However, to what extent these two processes are related and through which binding partners and/or enzymatic processes remains to be elucidated (Le Guen and De Zeeuw, Funct. Neurol., 2010); (2) Lonart et al. (Cell, 2003) found that RIM1α-Ser413 is phosphorylated by PKA, which is required for presynaptic LTP. However, mice with a Ser413 mutation exhibit normal presynaptic LTP in the cerebellum and the hippocampus (Kaeser et al., PNAS, 2008; Yang and Calakos, J Neurosci, 2010), questioning the central roles of RIM1α-Ser413 and PKA in presynaptic LTP and how phosphorylation of RIM1α comes about… Here, we show that, upon cAMP activation, EPAC (a cAMP target protein) induces PKCε-dependent threonine phosphorylation of RIM1α and that presynaptic parallel fiber-Purkinje cell LTP depends on an EPAC-PKCε module.” Therefore to be frank, our decision to study EPAC in PCs was based on the prevailing scientific questions.

Although three reviewers agreed to the conceptual advance and found it interesting, they finally rejected the submission with two major reasons: (1) the knockout was not cell specific (EPAC1 and EPAC2 double knockout was used in that submission); (2) there was no behavioral study. To answer these critiques, we endeavored to create 3 conditioned knockout mice (*Epac1*-cKO, *Epac2*-cKO, *and Prcke2*-cKO) during the very hard pandemic time. Furthermore, we built VOR setup and re-conducted all electrophysiological and behavioral studies.

The above rationale was added in the introduction (Line 52). Martín et al. (2020) paper was also added (Line 63) “which is in line with previous work (Martín et al., 2020) showing β-adrenergic receptors/EPAC signaling modulates PF release using EPAC2 knockout mice”.

2. Figure 1C: The statement that ENACs bind to RIM would be more convincing if the authors provide a negative control showing that the assay does not detect another presynaptic protein that does not interact with RIM does not appear in the pulldown. That said, interactions between ENAC and RIM have been reported before, and those studies could be cited.

Thank you. We have cited previous work showing the interactions between

EPAC and RIM (Ferrero et al., 2013) (Lines 483). Our preliminary experiments have detected the interaction of EPACs with other several important transmitter release-related proteins, such as MUNC13, MUNC18, and synaptophysin. However, the immunoprecipitation assay revealed no interaction between EPACs and these proteins

(see Author response image 1).

**Author response image 1. sa2fig1:** 

3. Figure 2E, F: I have multiple concerns with this data. First, the RRP is calculated from EPSC trains collected evoked by only 40 stimuli. 40 stimuli are insufficient to reach a steady state at PF-PC synapses, which is evident in the example EPSC traces. Without reaching a steady state, it is impossible to estimate the resupply rate. Thus the RRP measurements are not accurate. Second: Without an accurate measure of the RRP, the initial release probability cannot be determined. Finally, although Line 166 states that the RRP was significantly reduced in the mutant mice, that data is not presented in the text of the figure. Instead, the figure shows the "cumulative EPSC", which is highly dependent on the number of presynaptic terminals that were stimulated. The average RRP sizes, measured by back extrapolation, actually appear to be very similar across genotypes. The authors should not make claims about release probability or RRP unless they repeat these experiments with longer stimulus trains that reach steady state amplitudes, and the RRP is normalized to the size of the initial EPSC in each experiment.

We thank the reviewer for this valuable feedback. Combined with the suggestion of the reviewer#3, we have redone the RRP and Pr experiments using a 100-Hz train (instead of 20-Hz which was used in the previous submission). This method has allowed us to have a much better estimation on RRP and Pr at PF-PC synapses, according to Thanawala and Regher (2016). The new results have been presented in new Figure 2E and 2F.

Question #1: in new experiments, we have paid particular attention to ensure that recorded EPSCs reached a steady-state level by giving 50 stimuli, so that the regression can be conducted using at least 20 data points for the back extrapolation.

Questions #2 and #3, we have created a linear fit from normalized steady-state

EPSCs and back-extrapolated the curve to the y-axis to obtain the RRP. Our results showed that RRP was similar between control and cKO mice (Math1-cre: 861 ± 113 vs *Epac1*^cKO^;*Epac2*^cKO^ 790 ± 101, *p* = 0.31; PKCε^f/f^ 764 ± 100 vs PKCε^cKO^ 728 ± 106, *p* = 0.40). Actually, there was also no difference in RRP between control and cKO mice in our previous submission, but we incorrectly wrote “repeated stimulation (20 Hz) revealed significant reductions in both RRP and Pr in *Epac1*^cKO^;*Epac2*^cKO^ (*Figure 2E*) and *Prkce*^cKO^ mice (*Figure 2F*)”. Here, “RRP” was added incorrectly, we meant that only “Pr” was reduced. In the revised figure, we add the statistics of RRP in Figure 2E and 2F.

Finally, we have found that the conditional knockout of either EPACs or PKCε produces significant decrease on Pr (Math1-cre 0.17 *vs* Math1-cre;EPAC1^cKO^EPAC2^cKO^ 0.11; PKCε^f/f^ 0.19 *vs* PKCε^cKO^ 0.12). These results have been added in the text and figure legend (Figure 2E and 2F), and corresponding methods have also been updated.

4. Line 240: "However, it remains unclear which downstream effector, EPAC or PKA (Cheng et al., 2008), is responsible for cAMP-induced potentiation." Martin et al. 2020 present compelling evidence that presynaptic LTP is driven by EPAC at the PF-PC synapse. This study is cited obliquely, and should not be ignored to enhance the novelty of the current study. For example, Line 433 understates the relevance of Martin to the current study: "EPAC1 and EPAC2 have been shown to be involved in synaptic release in the hippocampus and cerebellum".

We are sorry for these confusions. We have modified the paragraph in Line 283 to “Next, we wondered which downstream effector, EPAC or PKA (Cheng et al., 2008), is responsible for cAMP-induced potentiation. The role of PKA in presynaptic LTP has been contradicted by the studies showing that presynaptic LTP is intact when serine phosphorylation of RIM1 by PKA is interrupted (Kaeser et al., 2008; Yang and Calakos, 2010; also see Lonart et al., 2003). Moreover, Martín et al. (2020) showed that EPAC2 regulates synaptic release at PF synapses and is required for presynaptic PF-PC LTP. These findings inspired us to investigate whether perhaps the EPAC-PKCε module mediates cAMP-triggered EPSC potentiation”.

Both EPAC1 and EPAC2 are the effectors of cAMP, and widely present in the brain. In earlier studies, the lab of Prof. Youmin Lu has shown the roles of EPAC1 and EPAC2 in synaptic release in the hippocampus using double and single EPAC knockout mice (Yang et al., 2012; Zhao et al., 2013). Considering the reviewer’s suggestion, we have modified the sentence to “*Although early studies have shown that EPAC1 and EPAC2 are involved in synaptic release in the hippocampus and the cerebellum (Yang et al., 2012; Zhao et al., 2013), which was further strengthened by Martín et al., (2020), our finding that PKCε acts*…” (Line 486).

5. The rationale for proceeding immediately to EPAC1/2 DKOs was not clear, leaving the reader to guess the reason? Are EPAC1/2 redundant in the EPAC-Rim1 pathway, or does one isoform dominate in potentiating this synapse? Could the authors make it clear why all experiments were done with DKOs?

EPAC1 and EPAC2 are both homologous receptors of cAMP and share highly conserved cAMP-binding domains (Cheng et al., 2008). They both act on the downstream Rap1 signaling, and have significant cross-talk and redundant roles in many physiological processes (Cheng et al., 2008). Thus, we cannot exclude the possibility that they both work on Rim1 at PF synapses, plus previous studies in the hippocampus using EPAC1 or EPAC2 knockout mice (Yang et al., 2012; Zhao et al., 2013). It remains unclear whether EPAC1 or EPAC2 alone is sufficient and which one may dominate modulation of Rim1 phosphorylation. We thereby have added a paragraph in the discussion to explain the rationale for using EPAC1/EPAC double cKO mice “*EPAC1 and EPAC2 share highly conserved cAMP-binding domains, and have significant cross-talk and redundant roles in many physiological processes (Cheng et al., 2008). Thus, we simultaneously deleted EPAC1 and EPAC2 in the granule cells in the present work, leaving the question open whether one might dominate the modulation of Rim1α phosphorylation*” (Line 449)*.*

6. Figure 4: What concentration of KT5720 was used to inhibit PKA? The authors note on Line 441 that previous studies used 10 microM, a concentration that produces off-target effects, but I cannot find the concentration the authors used. From their experiments, the authors conclude that the EPAC-PKC module is dominant in driving forskolin-induced potentiation. However, an obvious experiment would have been to apply KT to the control animals (Math1-Cre). This would rule out the possibility that PKA is activated somewhere within the EPAC-PKC, and plays a major role in potentiation.

Here are our responses.

1) We apologize for the missing information. The concentration of KT5720 used was 3 μM, which is close to the IC50 of PKA. According to Murray (2008), this concentration should minimize KT5720 effects on irrelevant signaling molecules, including EPAC, but still act on PKA.

2) We showed that KT5720 further decreased the potentiation by forskolin in conditional EPAC1/2 or PKCε knockout mice, albeit its effect was rather limited (Figure 4A). This result was also depicted in our summary Figure 7—figure supplement 2 (dashed line). Therefore, we believe that the impact of the EPAC-PKC module is much more significant than that of PKA in cAMP signaling pathway, at least at PF-PC synapses.

3) Following the reviewer’s comment, we have conducted new experiments to investigate the effect of KT5720 on the forskolin-induced potentiation in Math1-Cre mice. We found that KT5720 inhibited the forskolin-induced potentiation around 15%, significantly different but much smaller than the effect of EPAC or PKCε ablation. The result has been presented in Figure 4—figure supplement 1 and text (Line 304).

Reference: Murray AJ. 2008. Pharmacological PKA inhibition: all may not be what it seems. *Science Signaling* 1:re4.

7. The authors hypothesize that PLC is involved in this pathway (line 93 and Figure 1). They might use the discussion to highlight which aspects of their signaling pathway remain to be experimentally verified in presynaptic LTP.

Great suggestion. We have added the point in the discussion “Similarly, it would be of interest to investigate the role of Rap1, the presynaptically expressed substrate of EPAC a (Yang et al., 2012), on RIM1α phosphorylation” (Line 452).

Reviewer #2 (Recommendations for the authors):The introduction part should elaborate more on the rationale behind the selection of EPAC to conduct the study of presynaptic plasticity.

Thank you very much for this suggestion. We actually initiated this study based on previous controversy, regarding whether PKA is involved in the cerebellar plasticity. We then found EPAC, another downstream molecule of cAMP, and considered it a good candidate to test.

In the revised manuscript, we have added our rationale “In particular, the function of cAMP-dependent protein kinase A (PKA) on transmission release has been the subject of debate. Lonart et al. (2003) found that RIM1α-Ser413 is phosphorylated by PKA, which is required for presynaptic LTP. However, the mice with dysfunctional RIM1α-Ser413 mutation exhibit normal presynaptic LTP in the cerebellum and the hippocampus (Kaeser et al., 2008; Yang and Calakos, 2010), questioning the role of RIM1α-Ser413 and PKA in presynaptic LTP. Thus, how RIM1α is activated during presynaptic plasticity needs to be revisited” (Line 52).

The activities of EPAC and PKCε required for presynaptic LTP are shown. However, the change of the basal synaptic properties due to the knockdown of two proteins is not shown, such as the input-output relation of evoked EPSC in the PF-PC synapse. Therefore, studying the protein during the normal condition may be of interest.

In the new experiment, we have conducted input-output relationship of evoked EPSCs. Our results showed that the amplitudes of evoked EPSCs with various stimulation intensities were reduced by presynaptic deletion of EPAC or PKCε. The data have been shown in Figure 2—figure supplement 1 and in the text “Furthermore, we examined the evoked PF-PC EPSCs with different stimulation intensities (3-15 μA) in control and mutant mice. Our results showed that presynaptic deletion of either Epac1/Epac2 or Prkce significantly decreased evoked EPSCs in response to all stimuli (Figure 2—figure supplement 1)” (Line 181).

The visual diagram (Figure 1G) of the authors' hypothesis on a process describing EPAC-mediated phosphorylation of RIM1 is a bit difficult to follow. Arrows may be indicated with more comprehensible signs/ words.

We are sorry for the confusion. We have updated the Figure 1G and legend to make the point clear.

Following the compensatory eye movement training, the deficit is more prominent in phase than gain. Mutants catch up the gain at the end of 5 days of training but not the phase. The authors may explain how the presynaptic LTP is more essential to phase in the discussion. In addition, the reason to choose VOR phase reversal learning other than OKR learning or VOR gain-up learning may be related to this discussion.

The reviewer is correct in that the phase difference is the only difference present at the end of our VOR training. However, we found that the gain was also different in multiple days in both EPAC-cKO and PKCε-cKO mice compared to their controls. VOR phase reversal training subject mice to multiple days of changing training stimuli, unlike VOR or OKR gain increase, and thereby tests different aspects of adaptation. It should be noted that the aim of VOR phase reversal training is to first decrease the gain, followed by an increase in phase. If the training is continued long enough, the gain will increase again, usually when the phase reaches the levels higher than 120 degrees. Therefore, the initial decrease of gain followed by the late-stage increase presumably underlies the absence of significant differences in gain between control and mutant groups in days 4 and 5. This does not imply that presynaptic LTP is more essential for the VOR phase than VOR gain, as the VOR gain decrease is affected during the first days of training.

This explanation has been included in the end of the manuscript “Although we observed only a difference in phase at the end of VOR phase reversal training, it should be noted that the gain was different on multiple days in both Rapgef3;Rapgef4-cKO and Prkce-cKO mice compared to their controls. VOR phase reversal training subjects to multiple days of changing training stimuli to test different aspects of adaptation. The first aim is to decrease the gain, followed by an increase in phase. Once the phase has increased above 120o, the gain will increase again (Wulff et al., 2009). Therefore, the initial decrease of gain followed by the late-stage increase presumably underlies the absence of differences in gain between control and mutant groups in days 4 and 5. This does not imply that presynaptic LTP is more essential for the phase than the gain, as VOR gain decrease is affected during the first day of training”.

In the discussion part, some more thoughts and scientific predictions of the limited role of EPAC and PKCε on postsynaptic plasticity should be stated.

We appreciate this comment. In our work, we used the mouse models with presynaptic deletion of EPAC or PKCε, which affects PF transmitter release but not postsynaptic plasticity.

In the classical theory, induction of postsynaptic PF-LTD requires massive Ca influx, mGluR1 activation, cPLA_2_α/COX2-PKCα and CaMKII activation, GluA2 trafficking and its regulators, and CB1 receptor activation and high NO production; the induction of postsynaptic PF-LTP requires low Ca influx, cPLA_2_α activation, low NO production, phosphatase activation, and GluA2 trafficking (Wang and Shen, Chapter: *Plasticity of the cerebellum*; in Essentials of Cerebellum and Cerebellar Disorders; 2023; Springer Nature Group; https://link.springer.com/book/10.1007/978-3-031-15070-8). Based on current studies, we speculate that EPAC or PKCε might be unrelated to these molecular processes.

Interestingly, Gutierrez-Castellanos et al. (2017) showed that EPAC might regulate GluA3 conductance and is essential to postsynaptic LTP. Since an inhibitor of EPAC was used in that study, the genetic ablation of EPAC in PCs would be needed to repeat the results. However, this study gives us a hint that postsynaptic EPAC or PKCε may regulate the conductance of AMPA subunits, and then regulate postsynaptic LTP and LTD.

We have added a paragraph in Line 519 “In addition, Gutierrez-Castellanos et al. (2017) showed that EPAC may regulate GluA3 conductance in PCs, suggesting that postsynaptic EPAC or PKCε may regulate the conductance of AMPA receptor subunits, and thereby postsynaptic LTP or LTD at PF-PC synapses”.

Overall, specific claims made by the authors were straightforward and supported by an ample amount of experimental results. This study has widened the spectrum of understanding of the molecular complexity of cerebellar motor learning by demonstrating the active contribution of presynaptic plasticity and its related signaling cascade.

Thank you for these positive comments.

Reviewer #3 (Recommendations for the authors):The conclusions often overstate the findings of the data. The language can be toned-down a without diminishing the importance of these findings.

We thank you for this important comment. We have modified our conclusions throughout the manuscript, also based on the comment of the other reviewer.

In the present form, I don't think the cumulative EPSC analysis is informative. The authors should repeat the analysis using the Elmqvist and Quastel (EQ) method (see Thanawala and Regher, 2016). This method is more reliable at low Pr and can provide (when combined with the Train method) a high and low bracket of possible Pr and RRP values. Ideally, the authors would redo these experiments with a higher stimulus frequency. It would also be helpful to show a simple PPR value for each genotype. This can be done without additional experiments.

Thanks for the very insightful comment. In the previous experiments, we measured RRP and Pr based on parameter taken from the work in the hippocampal CA1 neurons (He et al., 2019), which, in our opinion, is similar to PF-PC synapses concerning low release probability. We have carefully read Thanawala and Regher (2016) paper and compared different methods. While the performance of the EQ method is in general more reliable to estimate small RRP and low Pr, it relies on *p* to be constant throughout a stimulus train (Thanawala and Regher, 2016). Although *p* may be constant for the calyx of Held synapses they studied, it cannot be case for PF-PC synapses. Therefore, we decided to redo the estimations of RRP and Pr using 100-Hz train (previously 20-Hz train).

This method does not require constant *p* and allows us to have a better estimation on RRP and Pr at PF-PC synapses (Thanawala and Regher, 2016).

The new results have been presented in new Figure 2E and 2F. The PF-PC synapses were stimulated at the frequency of 100 Hz, and the artifacts were truncated and the EPSCs were aligned (Figure 2E and 2F). Note that the aim of this experiment was to investigate whether there is difference between control and cKO mice. Indeed, we found that the amplitudes of both EPSC_0_ and follow-up EPSCs were smaller in cKO mice, indicating that both the initial release and the replenishment are reduced by the conditional knockout o EPACs or PKCε. Compared to 20-Hz train, the 100-Hz train resulted in steady-state EPSCs brought EPSCs into steady state faster. We created linear fit from normalized steady-state EPSCs and back-extrapolated the curve to the y-axis to calculate Pr. Indeed, we found that the Pr value estimated from the 100-Hz train stimulus was significantly larger than that from the 20-Hz train, showing 0.17 (Math1-cre) and 0.19 (PKCε^f/f^) with 100-Hz, but 0.07 (Math1-cre) and 0.08 (PKCε^f/f^) in previous submission. This result was similar to Thanawala and Regher (2016), in which they claimed that the accuracy of estimation from a 100-Hz train is about three times of that from a 20-Hz train. Moreover, we found that the conditional knockout of either EPACs or PKCε produced significant decrease on Pr (Math1-cre 0.17 *vs* Math1-cre;EPAC1^cKO^EPAC2^cKO^ 0.11; PKCε^f/f^ 0.19 vs PKCε^cKO^ 0.12). These results have been added in the text and figure legend (Figure 2E and 2F), and corresponding methods have also been updated.

Slice conditions in LTP/LTD experiments (age, location, temperature, Ca^2+^ concentration) do not match in vivo conditions. These differences and potential changes in synaptic plasticity should be discussed.

To date, almost all PC plasticity in published work were recorded in young adult mice (< 1 month) and at room temperature, and most behavioral experiments were conducted around 2-3 months of age. To better answer the reviewer’s comment, we tried our best to redo the LTP experiments under the requested, alternative conditions (in 2-month-old mice, low Ca^2+^ or high recording temperature). Our new data show that, under these conditions, EPACs and PKCε are still needed for the induction of presynaptic PC-LTP (Figure 3—figure supplement 2-4).

In addition, we have tried to record PC EPSCs in the flocculus. Unfortunately, we found PC EPSCs there were quite unstable, which might be due to the more complex orientation of PCs and their innervations. We have discussed the reviewer’s comment in the revised manuscript “Second, presynaptic PF-PC LTP was performed in the cerebellar vermis in the present work, whereas VOR learning generally requires PC activity in the flocculus. Unfortunately, we found that PC-EPSCs in the flocculus were not suitable to record PC plasticity because they were unstable” (Line 557).

Is it possible to provide an estimate of what percent of synaptosomes arise from parallel fibers versus mossy fibers (which also express VGLUT1) based on the relative number of each synapse? This would help the interpretation of the synaptosome data.

We have performed synaptosome staining vGluT1/vGluT2, EAAT4 and bassoon to identify PF-PC synapses (vGluT1^+^EAAT4+) or CF-PC (vGluT2^+^EAAT4+) synapses. Our staining results showed that PF-PC synapses covered 88.8% of the total and CF-PC synapses covered 7.5% of the total. Thus, we estimated the number of mossy fiber synapses to be less than 3.7%, which would not affect our conclusion. These results have been presented in Figure 1—figure supplement 1.